# A charged diatomic triple-bonded U≡N species trapped in $C_{82}$ fullerene cages

Qingyu Meng[1,6], Laura Abella[2,6], Yang-Rong Yao[3], Dumitru-Claudiu Sergentu [4], Wei Yang[1], Xinye Liu[1], Jiaxin Zhuang[1], Luis Echegoyen[5], Jochen Autschbach [2] ✉ & Ning Chen [1] ✉

Actinide diatomic molecules are ideal models to study elusive actinide multiple bonds, but most of these diatomic molecules have so far only been studied in solid inert gas matrices. Herein, we report a charged U≡N diatomic species captured in fullerene cages and stabilized by the U-fullerene coordination interaction. Two diatomic clusterfullerenes, viz. UN@$C_s$(6)-$C_{82}$ and UN@$C_2$(5)-$C_{82}$, were successfully synthesized and characterized. Crystallographic analysis reveals U-N bond lengths of 1.760(7) and 1.760(20) Å in UN@$C_s$(6)-$C_{82}$ and UN@$C_2$(5)-$C_{82}$. Moreover, U≡N was found to be immobilized and coordinated to the fullerene cages at 100 K but it rotates inside the cage at 273 K. Quantum-chemical calculations show a (UN)$^{2+}$@($C_{82}$)$^{2-}$ electronic structure with formal +5 oxidation state (f$^1$) of U and unambiguously demonstrate the presence of a U≡N bond in the clusterfullerenes. This study constitutes an approach to stabilize fundamentally important actinide multiply bonded species.

Fullerenes are known for their unique ability to encapsulate metal ions and clusters in their hollow interior. Clusterfullerenes, whose molecular structures are formed by the mutual stabilization between the entrapped metal clusters and the outer carbon cages, have become the most versatile and diverse category of endohedral metallofullerenes (EMFs) family[1]. Many of the entrapped clusters, including nitrides, carbides, oxides, sulfides, and cyanides, are otherwise unstable. Thus, besides their physical and chemical properties, cluster fullerenes also provide an ideal molecular model to study clusters that otherwise could not be prepared. Our recent studies showed that very diverse actinide clusters containing important actinide bonding motifs can be formed and stabilized inside the fullerene cages by electron transfer between the cluster and carbon cage and by the U-fullerene coordination. They can therefore be systematically characterized in the form of molecular compounds[1]. For example, a long-sought axial U=C bond with the shortest U-C bond distances discovered so far, was found to be stabilized in the form of an encapsulated U=C=U cluster in an actinide clusterfullerene, $U_2C@I_h$(7)-$C_{80}$[2]. Subsequent studies further

revealed the variety of the actinide clusterfullerene families, with the successful synthesis and characterization of $U_2C_2@I_h$(7)-$C_{80}$ and UCN@$C_s$(6)-$C_{82}$[3,4]. Encapsulated $U_2C_2$, which presents two U bridged by C≡C triple bond, and a triangular UCN cluster, which features $\eta^2$ (side-on) coordination of U by cyanide, show novel bonding motifs for U, broadening our understanding of the chemical properties of the actinide elements.

Covalent bonding with the 5f and 6d orbitals in actinide−ligand multiple bonds has been intensively studied, but remains incompletely understood both experimentally and theoretically[5]. The understanding of these bonding motifs is relevant for developing advanced nuclear fuel and managing radioactive waste. In particular, uranium nitrides have potential applications as nuclear fuel due to their high melting point and thermal conductivity[6]. Thus, a full understanding of U-N multiple bonding is essential for the future applications of uranium nitride compounds. From the aspect of synthesis, the terminal U-N multiple bond is very challenging because bond polarity is

[1]College of Chemistry, Chemical Engineering and Materials Science, and State Key Laboratory of Radiation Medicine and Protection, Soochow University, Suzhou, Jiangsu 215123, P. R. China. [2]Department of Chemistry, University at Buffalo, State University of New York. Natural Sciences Complex, Buffalo, NY 14260-3000, USA. [3]Department of Materials Science and Engineering, University of Science and Technology of China, Hefei 230026, P. R. China. [4]A.I. Cuza University of Iași, RA-03 Laboratory (RECENT AIR), Iași 700506, Romania. [5]Department of Chemistry, University of Texas at El Paso, 500 W University Avenue, El Paso, Texas 79968, USA. [6]These authors contributed equally: Qingyu Meng, Laura Abella. ✉e-mail: jochena@buffalo.edu; chenning@suda.edu.cn

stronger with actinides than with transition metals. This makes terminal actinide–ligand linkages difficult to stabilize, compared to bridging multiple-bond groups[7]. To date, only two classes of compounds containing terminal uranium nitrides were reported. In 2012, Liddle and co-workers reported the synthesis and characterization of a terminal uranium nitride complex, [UN(Tren$^{TIPS}$)] [Na(12C4)$_2$][8], which was subsequently reduced to give a uranium(VI)–nitride triple bond in [UN(Tren$^{TIPS}$)][9]. Move recently, Mazzanti obtained [NBu$_4$][U(OSi(O$^t$Bu)$_3$)$_4$(N)] with a terminal U≡N by photochemical synthesis[10].

In fact, U≡N bonding motifs were initially found in small molecules prepared by nitrogen discharging or laser-ablation. The U≡N bond was first prepared and identified by Green and Reedy in 1976 from a nitrogen discharge in the presence of uranium metal and then studied by several groups[11,12]. Numerous terminal uranium nitride molecular species, such as N≡U≡N, (NUN)(NN)$_x$, U≡NF$_3$ and N≡U=N-H have also been reported[12–15]. However, these molecular species have only been studied using cold matrix-isolation methods as well as quantum-chemical calculations. Thus, crystallographic characterization of these molecular species remains elusive. As mentioned, our previous studies have demonstrated the ability of fullerenes to encapsulate variable novel uranium bonding motifs. Therefore, we wondered whether fullerene cages could capture and stabilize the aforementioned small actinide molecular species. And if so, what kind of interactions would be observed?

Herein, we report the synthesis and characterization of two diatomic clusterfullerenes, UN@$C_s$(6)-C$_{82}$ and UN@$C_2$(5)-C$_{82}$. For this family of endohedral fullerenes, a charged diatomic species with a U≡N triple bond is captured and stabilized by the C$_{82}$ fullerene cage isomers. X-ray single-crystal diffraction shows a very short U-N distance of 1.760(7) Å and 1.760(20) Å in UN@$C_s$(6)-C$_{82}$ and UN@$C_2$(5)-C$_{82}$, comparable to those predicted for the UN cluster under matrix-isolation conditions. Variable-temperature X-ray single-crystal diffraction (VT-SC-XRD) reveals a unique host-guest interaction between UN and the fullerene cage at different temperatures. Calculations show that UN transfers two electrons to the $C_2$(5)-C$_{82}$ or $C_s$(6)-C$_{82}$ cages, resulting in a (UN)$^{2+}$@(C$_{82}$)$^{2-}$ electronic structure. Density functional and wavefunction calculations unambiguously show U-N bond orders in agreement with a genuine triple bond.

## Result

### Synthesis and isolation of UN@$C_s$(6)-C$_{82}$ and UN@$C_2$(5)-C$_{82}$

A modified Krätschmer-Huffman DC arc-discharge method was used to synthesize UN@$C_s$(6)-C$_{82}$ and UN@$C_2$(5)-C$_{82}$. U$_3$O$_8$ powder and graphite powder (molar ratio of U/C=1:30) were mixed and then deposited into hollow graphite rods, and reacted in the arcing chamber under a 200 Torr He and 4 Torr N$_2$ atmosphere. The collected carbon soot containing uranium-based metallofullerenes was extracted with CS$_2$ for 12 h. The UN@$C_s$(6)-C$_{82}$ and UN@$C_2$(5)-C$_{82}$ were separated and purified by multi-step HPLC, and the separation process was monitored by mass spectrometry (Supplementary Figs. 1, 2). It is noteworthy that UO@C$_{82}$ is also observed during the HPLC separation process, possibly due to the leak of air into arcing chamber, but was removed during the purification processes (Supplementary Fig. 3a, b). The purity of the samples was confirmed by the observation of a single peak HPLC chromatography. Furthermore, the high resolution mass spectrum of the final purified sample also shows that the isotopic distribution of the samples obtained experimentally is consistent with the theoretical isotopic distribution of UN@C$_{82}$, excluding the existence of UC@C$_{82}$ or UO@C$_{82}$ (Supplementary Fig. 3c, d).

### Molecular and electronic structures of UN@C$_{82}$

Black block cocrystals of UN@C$_{82}$ with Ni$^{II}$(OEP) (OEP = octaethylporphyrin dianion) were obtained by slow diffusion of a

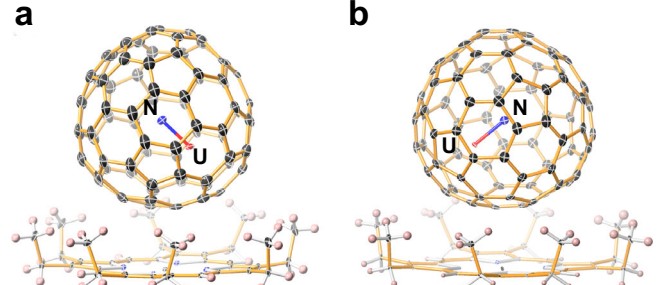

**Fig. 1 | ORTEP drawing of UN@$C_2$(5)-C$_{82}$·[Ni$^{II}$(OEP)] and UN@$C_s$(6)-C$_{82}$·[Ni$^{II}$(OEP)] with 20% thermal ellipsoids. a** UN@$C_2$(5)-C$_{82}$·[Ni$^{II}$(OEP)]. **b** UN@$C_s$(6)-C$_{82}$·[Ni$^{II}$(OEP)]. Only the major U site is shown. For clarity, the solvent molecules and minor metal sites are omitted.

benzene solution of [Ni$^{II}$(OEP)] into a CS$_2$ solution of UN@C$_{82}$ and characterized by single-crystal XRD analysis.

Figure 1 shows the molecular structures of UN@$C_2$(5)-C$_{82}$·[Ni$^{II}$(OEP)] and UN@$C_s$(6)-C$_{82}$·[Ni$^{II}$(OEP)] in space groups of $C2/m$ and $C2/c$, respectively. For UN@$C_2$(5)-C$_{82}$, the crystallographic results show two orientations of the fullerene molecule with equal occupancy of 0.5. These two orientations are symmetrical through the molecular crystallographic mirror (Supplementary Fig. 4). The major U position for UN@$C_2$(5)-C$_{82}$, with fractional occupancy of 0.312, is situated over the intersection of two hexagons and a pentagon, identical to that for the previously reported U@$C_2$(5)-C$_{82}$ (Supplementary Fig. 5)[16]. Theoretical calculations also show that the U1 site has the lowest relative energy (Supplementary Fig. 10). Thus, the optimal UN cluster orientation can be accurately determined, as shown in Fig. 1a. The mirror-related counterpart and other minor disordered U sites are shown in Supplementary Fig. 5. For UN@$C_s$(6)-C$_{82}$, the fullerene cage displays only one orientation. The U atom shows several disordered positions, of which U1(0.6442) is the dominant one. The other less-occupied disordered sites are shown in Supplementary Fig. 6. In contrast, the N atom is fully ordered in the center of the fullerene cage. The distances between the metal and the nearest six carbons on the fullerene cage in $C_s$(6)-C$_{82}$ and $C_2$(5)-C$_{82}$ are 2.478(15)−2.861(22) Å and 2.503(7)−2.785(7) Å (Supplementary Tables 1, 2), which agree with the theoretical calculations (Supplementary Table 3, 2.503-2.634 Å for $C_2$(5)-C$_{82}$ and 2.476−2.631 Å for $C_s$(6)-C$_{82}$). In addition, these distances are also close to the U-Cp distances in U(V) organometallic compounds, such as 2.723(3)−2.830(3) Å in (Cp$^{iPr4}$)$_2$U(μ-N)B(C$_6$F$_5$)$_3$[17] and 2.718(7)−2.866(7) in {U[η$^8$-C$_8$H$_6$(1,4-Si-($^i$Pr)$_3$)$_2$](Cp*)(NSiMe$_3$)}[18]. This indicates that the interaction between the fullerene cage and U ion likely bears some resemblance to the coordination between the metal and the cyclopentadienyl group in organometallic compounds.

UN@$C_s$(6)-C$_{82}$ and UN@$C_2$(5)-C$_{82}$ represent examples of an encapsulated charged diatomic species, the simplest encapsulated specie obtained for all endohedral clusterfullerenes[19]. The U-N distances corresponding to the major configurations in UN@$C_s$(6)-C$_{82}$ and UN@$C_2$(5)-C$_{82}$ are measured as 1.760(7) and 1.760(20) Å, much shorter than the U=N bonds (1.943(3)/2.058(3) Å) in U$_2$N@$I_h$(7)-C$_{80}$[20], but close to the terminal U≡N bond lengths observed in coordination compounds[8–10,21–24]. The U-N distances in the other configuration corresponding to its minor U sites are 1.681(7)−1.820(9) Å (U2 (0.1903), U3 (0.1087) and U4 (0.0566) for UN@$C_s$(6)-C$_{82}$) and 1.705(20) Å (U2 (0.188) for UN@$C_2$(5)-C$_{82}$) (Supplementary Figs. 4 and 6 and Supplementary Tables 14, 15), all of which are within the bond length range of a U≡N triple bond. Thus, based on distance, the U-N bonds in both UN@$C_s$(6)-C$_{82}$ and UN@$C_2$(5)-C$_{82}$ can be assigned as terminal U≡N bonds. For comparison, most known encaged clusters reported to date feature single bonds between metal and non-metal atoms[19]. So far, only M$_2$TiC@C$_{80}$ (M=Sc and Lu), USc$_2$C@C$_{80}$ and U$_2$C@C$_{80}$ were found to

**Table 1 | Crystal data of U≡N bond lengths and calculated U≡N bond lengths in gas phase molecules**

| Samples | U-N bond lengths | Reference |
|---|---|---|
| UN@$C_s$(6)-$C_{82}$ | 1.760(7) Å | This work |
| UN@$C_2$(5)-$C_{82}$ | 1.760(20) Å | This work |
| [U(N)(Tren$^{TIPS}$)] | 1.799(7) Å | 9 |
| [U(OSi(O$^t$Bu)$_3$)$_4$(N)]$^-$ | 1.769(2) Å | 10 |
| [(NH$_3$)$_8$U($\mu$-N)Cl$_2$(NH$_3$)$_3$U($\mu$-N)U(NH$_3$)$_8$]Cl$_6$$^{6+}$ | 1.853(5)/1.834(5) Å | 23 |
| U≡N | 1.746 Å | 28 |
| N≡UF$_3$ | 1.759 Å | 14 |
| N≡UNH | 1.742 Å | 15 |

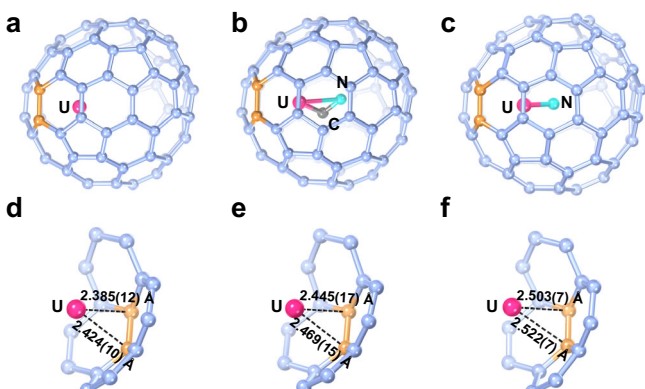

**Fig. 2 | Structures and distances between U and the fullerene cage. a**, **d** U@$C_s$(6)-$C_{82}$, **b**, **e** UCN@$C_s$(6)-$C_{82}$, and **c**, **f** UN@$C_s$(6)-$C_{82}$.

have M=C bonds[2,25–27]. Here, in UN@$C_s$(6)-$C_{82}$ and UN@$C_2$(5)-$C_{82}$, U≡N bonds are trapped inside fullerene cages. Moreover, as shown in Table 1, these U≡N distance are shorter or approximately equal to the observed U≡N bond lengths for molecular compounds, such as[U(N) (Tren$^{TIPS}$)] (1.799(7) Å)[9], [U(OSi(O$^t$Bu)$_4$)(N)]$^-$ (1.769(2) Å)[10] and [(NH$_3$)$_8$U($\mu$-N)Cl$_2$(NH$_3$)$_3$U($\mu$-N)U(NH$_3$)$_8$]Cl$_6$$^{6+}$ (1.853(5)/1.834(5) Å)[23], and very close to those calculated bond lengths in gas phase molecules and clusters studied only by matrix-isolation, such as U≡N (1.746 Å)[28], N≡UF$_3$ (1.759 Å)[14], and N≡UNH (1.742 Å)[15]. These short UN distances are similar to the short UO bonds in the simplest U(VI) uranyl complexes, such as U(VI)O$_2$($^{tBu}$acnac)$_2$ which has a UO bond with bond length of 1.770(3) Å[29].

Theoretical calculations were employed to determine the electronic structure and bonding for UN@$C_s$(6)-$C_{82}$ and UN@$C_2$(5)-$C_{82}$. If not noted otherwise, computational data discussed hereafter were obtained from DFT calculations with the ZORA[30] all-electron relativistic Hamiltonian, matching triple-zeta polarized Slater-type orbital (STO) basis sets[31,32], the Perdew-Burke-Ernzerhof (PBE) functional[33], and semiempirical dispersion corrections (D3)[34,35], as implemented in the ADF package[36]. Consistent results for the ground state spin-multiplicity and structural parameters were obtained with other functionals (Supplementary Tables 8, 9). Likewise, spin state energetics and structural parameters consistent with ADF results were obtained using the G16 program[37,38] with Gaussian-type orbital (GTO) basis sets and effective core potentials to mimic relativistic effects, as seen in Supplementary Tables 10, 11. The latter approach was primarily used to calculate Raman spectra. Additional computational details are provided at the end of the article. Starting from the X-ray coordinates, the molecular structures of UN@$C_s$(6)-$C_{82}$ and UN@$C_2$(5)-$C_{82}$ were optimized for doublet and quartet spin states. The results are compiled in Supplementary Table 3. Note that $C_s$(6)-$C_{82}$ and $C_2$(5)-$C_{82}$ are related by a Stone-Wales transformation (SWT). The corresponding $C_2$ unit involved in this rearrangement is highlighted in yellow for isomer 5 in Supplementary Fig. 7. The spin-doublet ground state UN@$C_s$(6)-$C_{82}$ geometry is the most stable structure, followed by the spin-doublet UN@$C_2$(5)-$C_{82}$ geometry at 5.3 kcal·mol$^{-1}$ higher in energy. For both isomers, the spin-quartet equilibrium geometries are found at 17-18 kcal·mol$^{-1}$ higher in energy than those for the spin-doublet structures (xyz coordinates of optimized UN@$C_s$(6)-$C_{82}$ and UN@$C_2$(5)-$C_{82}$ structures are shown in Supplementary Tables 18, 19). Note that the experimental and calculated U-N distances are in very good agreement for both isomers 5 and 6. The closest U-C cage contacts are in the range of 2.476–2.634 Å, which is in agreement with other U cluster fullerenes such as U$_2$C$_2$@$I_h$(7)-$C_{80}$, U$_2$C@$I_h$(7)-$C_{80}$, U$_2$@$I_h$(7)-$C_{80}$, and UCN@$C_s$(6)-$C_{82}$[2–4,39]. Different orientations of the UN$^{2+}$ cluster within the $C_2$(5)-$C_{82}$ and $C_s$(6)-$C_{82}$ cages were also considered and explored for the spin-doublet configurations (Supplementary Figs. 10, 11 and Supplementary Tables 12, 13). They are higher in energy than structures optimized from the X-ray coordinates.

Calculations show that the encapsulated UN cluster transfers two electrons to the $C_2$(5)-$C_{82}$ or $C_s$(6)-$C_{82}$ cage, resulting in (UN)$^{2+}$@($C_{82}$)$^{2-}$ electronic structures. Therefore, the U center attains the formal +5 oxidation state (f$^1$). The system adopts a spin-doublet ground state with the unpaired electron localized at the U(V) center. Molecular orbital (MO) diagrams for the ground spin-doublet states of UN@$C_2$(5)-$C_{82}$ and UN@$C_s$(6)-$C_{82}$ are shown in Supplementary Fig. 12. The spin density (SD) and the U Mulliken spin populations (0.8 for isomer 5 and 0.5 for isomer 6), computed for the spin-doublet state, confirm that the unpaired electron is mainly localized at the metal center (Supplementary Fig. 7b). The nature of the covalent U-N interactions is discussed later.

DFT calculations were also performed for isolated UN and UN$^{2+}$. The molecular orbital diagrams and the corresponding spin densities for UN and UN$^{2+}$ are presented in Supplementary Figs. 15, 16. For neutral UN, a spin-quartet state is the lowest in energy, whereas for the isolated UN$^{2+}$ it is a spin-doublet. The DFT-optimized structure of isolated neutral UN shows a bond distance of 1.756 Å, while for the isolated dication UN$^{2+}$ the distance is shorter, 1.707 Å. The latter distance is somewhat shorter than that observed for the UN cluster fullerenes, which indicates that the strengths of the metal-ligand interactions and the metal-cage interactions are correlated. This is corroborated by the metal-cage distances, as discussed next.

The successful synthesis of UN@$C_s$(6)-$C_{82}$ provides a rare chance to study the metal-cage interaction of three structurally closely related actinide endohedral fullerenes, i.e. U@$C_s$(6)-$C_{82}$, UN@$C_s$(6)-$C_{82}$ and UCN@$C_s$(6)-$C_{82}$, which share the same $C_s$(6)-$C_{82}$ cage. The crystallographic analysis shows that, despite the completely different bonding structures of the three endo-units, surprisingly, the U ion position inside $C_s$(6)-$C_{82}$ is almost identical for the three molecules, which locates in the symmetric plane of UN@$C_s$(6)-$C_{82}$ and is close to a [6, 6] bond surrounded by three hexagons and a pentagon, as shown in Fig. 2. This identical U position enables us to assess the metal/cluster-cage interaction from the distance between the U and the cage carbons. These distances can be obtained by measuring the distances between the U atom and the [6, 6] bond, which clearly shows that the U-cage distances decrease in the sequence of UN@$C_s$(6)-$C_{82}$ (2.522(7)/ 2.503(7) Å), UCN@$C_s$(6)-$C_{82}$ (2.445(17)/2.469(15) Å) and U@$C_s$(6)-$C_{82}$ (2.385(12)/2.424(10) Å), indicating that the metal-cage interactions correlate with the differences in the bonding of the endo-units. As the metal-ligand bond strength in the encapsulated uranium cluster increases, the uranium-cage interaction is correspondingly weakened. Particularly, the interaction between the U atom and the fullerene cage is significantly weakened in the presence of the robust U≡N bond, leading to the longest metal-cage contacts among the three uranium-based EMFs. In fact, this U-cage distance is also the longest among all

the reported uranium-based EMFs (2.264(19) to 2.491(5) Å) thus far[2,3,16,20,39–42].

Variable-temperature single-crystal X-ray diffraction (VT-SC-XRD) was employed to study the dynamic behavior of the UN clusters inside the fullerene cages[43]. Supplementary Fig. 23 shows the crystal structures of UN@$C_s$(6)-C$_{82}$·[Ni$^{II}$(OEP)] measured at 100 K, 185 K, and 273 K (crystal data shown in Supplementary Table 6). The thermal vibration of all the atoms in the crystal increases significantly as the temperature rises, which can be observed by the enlarged thermal ellipsoids, indicating their temperature-dependent dynamic behaviors. The occupancy of the major site of U decreases as the temperature increases, and new disordered sites appear (detailed information of metal site occupation is listed in Supplementary Table 4). Interestingly, as summarized in Supplementary Table 14, even at the higher temperature of 273 K, the UN distances are at ca. 1.720(20)−1.801(10) Å, still indicative of a U≡N triple bond. This suggests that the increased temperature does not change the bonding structure of the UN cluster. Moreover, as shown in Fig. 3, at 185 K and 273 K, the U ion appears to "rotate" around the Ni···N axis. In other words, at higher temperatures the U atom samples alternate binding sites inside the cage while keeping their bond with N. The VT-SC-XCD characterization was also performed for UN@$C_2$(5)-C$_{82}$ at 100, 185 and 273 K (Supplementary Fig. 9 and Sup-

plementary Table 7), and the UN cluster shows a similar dislocation pattern as that for UN@$C_s$(6)-C$_{82}$. This study reveals an interesting interaction between the encaged UN cluster and the host carbon cage: At 100 K, the U≡N bonding motif is essentially immobilized inside the fullerene cages, which behave like a ligand coordinating to the U≡N unit. At higher temperature, however, UN is more mobile and the uranium samples different sites inside the fullerene cage.

## Spectroscopic characterization

The UV-vis-NIR spectra of the two isomers of UN@C$_{82}$ are shown in Fig. 4a. The absorption features of the two isomers of UN@C$_{82}$ are dominated by the π→π* excitation of their carbon π-system, commonly known for other reported endohedral fullerenes[44]. For UN@$C_s$(6)-C$_{82}$, there is a shoulder peak near 500 nm, and two broad slightly structured peaks at 1000-1400 nm and 600-800 nm, similar to the spectrum of UCN@$C_s$(6)-C$_{82}$ and the one previously reported for TbCN@$C_s$(6)-C$_{82}$[45]. On the other hand, UN@$C_2$(5)-C$_{82}$ shows a different absorption pattern from UN@$C_s$(6)-C$_{82}$, with two well-defined peaks at 772 and 1050 nm, resembling that of TbCN@$C_2$(5)-C$_{82}$[46]. This indicates similar isomeric structures and electronic transfer between UN@$C_2$(5)-C$_{82}$ and TbCN@$C_2$(5)-C$_{82}$, and between UN@$C_s$(6)-C$_{82}$ and TbCN@$C_s$(6)-C$_{82}$, respectively. These results are consistent with the computational results for [UN]$^{2+}$@C$_{82}$$^{2-}$ (both TbCN@$C_2$(5)-C$_{82}$ and TbCN@$C_s$(6)-C$_{82}$ have two electron cluster-to-cage electron transfer) and the crystallographic assignments of their different isomeric structures of $C_2$(5)-C$_{82}$ and $C_s$(6)-C$_{82}$.

UN@$C_s$(6)-C$_{82}$ and UN@$C_2$(5)-C$_{82}$ were further characterized by low-energy Raman and FTIR spectroscopy. Sharp peaks at 114 and 113 cm$^{-1}$ were observed in the Raman spectra of UN@$C_s$(6)-C$_{82}$ (Fig. 4e) and UN@$C_2$(5)-C$_{82}$ (Fig. 4c), respectively, which agree well with a 109 cm$^{-1}$ mode predicted by the Raman spectral calculations for both isomers, from UN wagging inside the cage (Fig. 4d, f). Notable metal-cage vibrational modes were also found at 150 and 149 cm$^{-1}$ for UN@$C_s$(6)-C$_{82}$ and UN@$C_2$(5)-C$_{82}$, respectively, similar to those previously reported for U-based clusterfullerenes, such as USc$_2$C@$I_h$(7)-C$_{80}$ (146 cm$^{-1}$)[25] and U$_2$C@$I_h$(7)-C$_{80}$ (148 cm$^{-1}$)[2]. In addition, theoretical calculations determined that the vibrational peaks for UN@$C_s$(6)-C$_{82}$

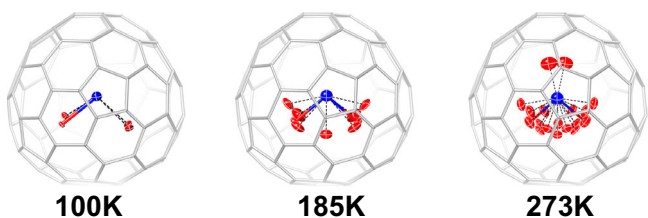

**100K** **185K** **273K**

**Fig. 3 | Molecular structure of UN@$C_s$(6)-C$_{82}$ measured with single-crystal X-ray diffraction at variable temperatures from 100 K to 273 K.** The displacement parameters are shown at the 20% probability level for the encapsulated UN cluster. The structures are drawn from the chosen specific direction of the crystal to compare the dynamics of the UN@$C_s$(6)-C$_{82}$. Color code: blue for N, and red for U.

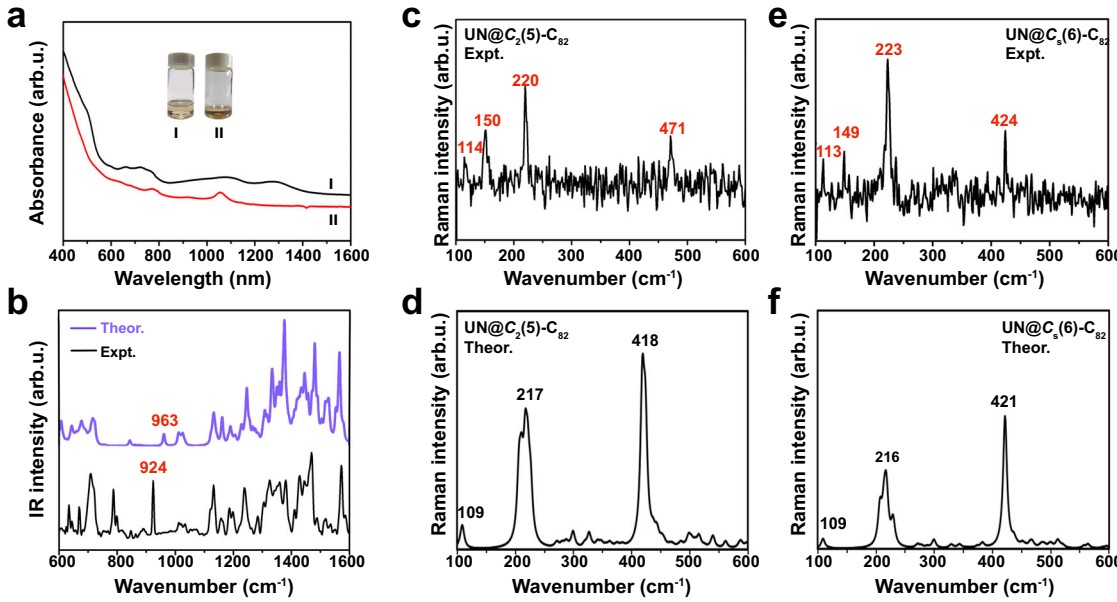

**Fig. 4 | Spectroscopic characterization of UN@$C_s$(6)-C$_{82}$ and UN@$C_2$(5)-C$_{82}$. a** UV−vis−NIR spectra of UN@$C_s$(6)-C$_{82}$ (I) and UN@$C_2$(5)-C$_{82}$ (II). **b** FTIR spectra of UN@$C_2$(5)-C$_{82}$. **c, d** Experimental Raman spectra of UN@$C_2$(5)-C$_{82}$ and theoretical

simulations. **e, f** Experimental Raman spectra of UN@$C_s$(6)-C$_{82}$ and theoretical simulations. Source data are provided as a Source Data file.

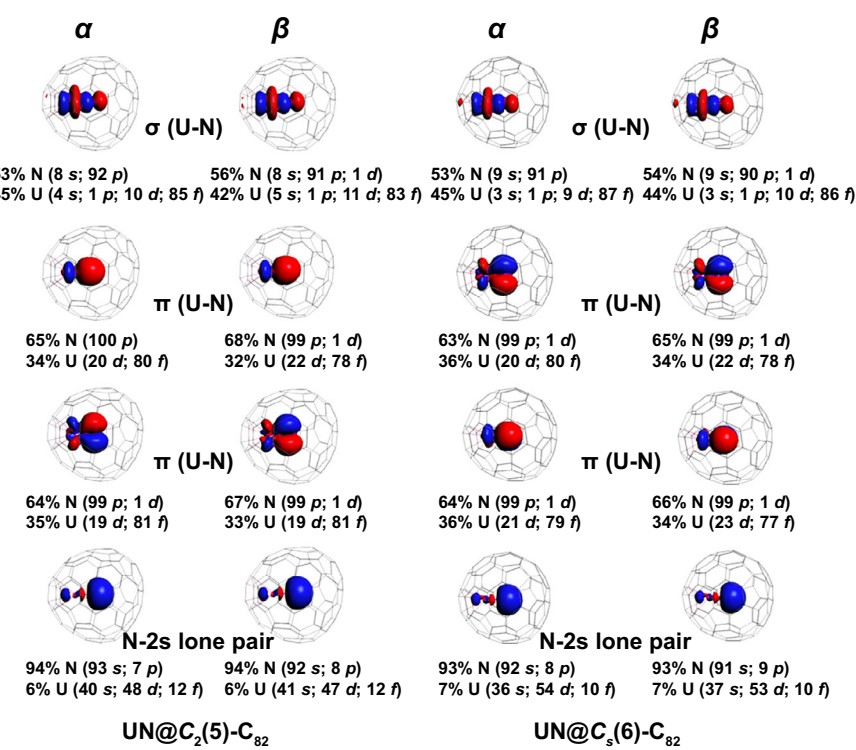

**Fig. 5 | NLMOs and atomic orbital %-weights for the U≡N triple bond.** NLMOs (±0.03 a.u. isosurfaces) and atomic orbital %-weights for the spin-doublet ground states of UN@$C_2$(5)-$C_{82}$ (left) and UN@$C_s$(6)-$C_{82}$ (right). Alpha ($\alpha$)- and beta ($\beta$)-spin orbitals are plotted separately.

and UN@$C_2$(5)-$C_{82}$, in the range of 200-500 cm$^{-1}$, can be assigned to cage vibrational modes (Supplementary Figs. 21, 22). In the FTIR spectrum of UN@$C_2$(5)-$C_{82}$ (Fig. 4b), the UN stretching mode can be assigned to the major peak of 924 cm$^{-1}$, which is comparable to the UN vibrational peak at 936 cm$^{-1}$ for [UN(Tren$^{TIPS}$)][Na(12C4)$_2$]$^8$[8]. In addition, the FTIR spectrum above 1000 cm$^{-1}$ corresponds to the vibrations of the carbon cage, reproduced well by the theoretical calculation.

Attempts to further resolve the electronic structure using EPR spectroscopy were unsuccessful, as no clearly defined signal was observed at 4 K (Supplementary Fig. 24). This behavior is supported by wavefunction theory (WFT) calculations (see Computational Details) for an isolated UN$^{2+}$ diatomic, $d$(U-N) = 1.707 Å, which predict a $J_z$ = 5/2 ground state Kramers doublet (Supplementary Table 17) characterized by an axial $g$ tensor with $g_{||}$ = $g_z$ = 4.19 and $g_\perp$ = $g_x$ = $g_y$ = 0. These values resemble those[22] calculated for a UN$^{2+}$ diatomic with d(U-N) = 1.84 Å ($g_z$ = 4.20, $g_x$ = $g_y$ = 0), and both are very close to the expected values for a $|J, J_z\rangle = |5/2, \pm5/2\rangle$, namely $g_z$ = 4.29 and $g_x$ = $g_y$ = 0. Similar WFT calculations for an isolated UN diatomic predicted a GS Kramers doublet with $J_z$ = 7/2 (Supplementary Table 16) and axial g tensor with $g_z$ = 3.99 and $g_x$ = $g_y$ = 0, which is an example case for a pure crystal field GS with $L_z$ = ±5, $S_z$ = ∓3/2, and $J_z$ = ±7/2 characterized by $g_z$ = 4.00 and $g_x$ = $g_y$ = 0. Absence of EPR signals were also reported for (MeC$_5$H$_4$)$_3$UNR compounds where the local $C_{3v}$ symmetry around the metal center renders nil values for $g_x$ and $g_y$[47]. The ESR signal corresponding to U(V) was also not observed in U$_2$C@C$_{80}$[2].

**Computational study of molecular bonding in UN@$C_2$(5)-$C_{82}$ and UN@$C_s$(6)-$C_{82}$**

DFT-based natural localized molecular orbital (NLMO)[48,49] bonding analyses for UN@$C_2$(5)-$C_{82}$ and UN@$C_s$(6)-$C_{82}$ demonstrate overall similar chemical bonding within the UN fragment. For both systems, there are three doubly occupied two-center NLMOs, one σ, and two π, describing the triple bond of UN$^{2+}$ (Fig. 5). For both isomers, the corresponding σ NLMO has about 53% total weight from N 2s-2p hybrids and 45% weight from U-based 6d (10%)–5f (86%) hybrids. For the π

NLMOs, the U weights are 34−36%, while for N they are 63-65%. The N-2s lone pair delocalizes toward U, with a modest contribution from the U center of 6-7%. The bonding orbitals are polarized toward N, as one would expect based on the high electronegativity of nitrogen. However, the combined weights of uranium atomic orbitals in these bonds, ranging from 34 to 45%, are large, such that the interaction between U and N can be classified as a genuine triple bond. Namely, compared to previous works on $^{15}$N bonds with thorium and related systems with U-C bonds, the UN bond in the clusterfullerene appears remarkably weakly polarized[7,50−52].

NLMO/DFT bonding analyses of UN and UN$^{2+}$ were also performed, showing that the corresponding analysis of UN$^{2+}$ gives very similar results as for UN@$C_2$(5)-$C_{82}$ and UN@$C_s$(6)-$C_{82}$. There are three two-center NLMOs, one σ and two π, describing the triple bond of UN$^{2+}$ (Supplementary Fig. 17). The σ NLMO has about 57% total weight from N 2s-2p hybrids and 43% weight from U-based 6d-5f hybrids. For the π NLMOs, the U weights are 40%. For UN$^{2+}$, there is considerably more f character in the triple bond than for neutral UN (Supplementary Fig. 18), in agreement with bonding analyses performed with the help of multiconfiguration wavefunction calculations (vide infra).

Some NLMOs that are centered mainly on the carbon cage have noticeable U contributions (4−10% weight) and pronounced U 5f character. They are shown in Supplementary Figs. 13, 14, respectively, along with the NLMO representing the unpaired 5f electron. The NLMOs evidence an interplay of donation to uranium and 5f-to-cage back-donation. There is a bit more donation from the $\alpha$ than from the $\beta$ spin orbitals of C$_{82}$ to U, compensating for the delocalization of the formally nonbonding $\alpha$−spin 5f NLMO into the cage. The NLMO study for both isomers shows covalent metal-cage interactions that go along with a slightly elongated U-N distance inside the fullerene, compared to isolated UN$^{2+}$, as a result of a somewhat reduced π bond covalency.

Low-energy electronic states were calculated with complete active space (CAS) multiconfigurational wavefunction theory (WFT)[53], with and without spin-orbit (SO) coupling (SOC), and with or without treatment of the dynamic correlation by second-order perturbation

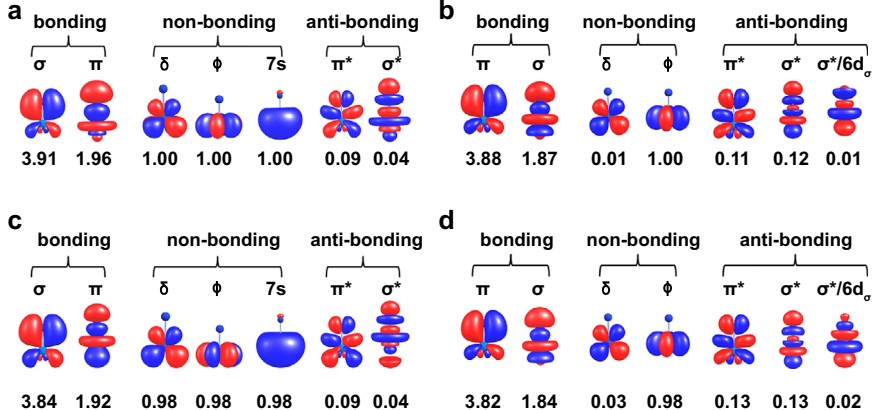

**Fig. 6 | Natural orbitals and populations from wavefunction calculations for UN and UN$^{2+}$.** Natural Orbital (NO) isosurfaces (±0.04 a.u.) and populations calculated for the $\Lambda = 3$ $^4$H ground state of UN without (**a**) and with PT2 treatment of dynamic correlation (**c**). NO isosurfaces (±0.04 a.u.) and populations calculated for the $\Lambda = 5$ $^2\Phi$ spin-free GS of UN$^{2+}$ without (**b**) and with the treatment of dynamic correlation (**d**). For degenerate NOs, one representative isosurface plot and the combined populations are shown.

theory (PT2) (see Computational Details). Given that DFT is known to have difficulties with the description of open-shell electronic states, WFT calculations were performed to confirm the bonding analysis derived from the DFT calculations. Key results are summarized in Supplementary Tables 16, 17 and the accompanying text in the Computational details section, for UN and UN$^{2+}$, respectively. Figure 6 displays the corresponding ground state (GS) natural orbitals (NOs) and populations. According to the PT2 calculations, UN has an effective bond order (EBO)[54] of 2.82 (Fig. 6a, c, determined from bonding minus anti-bonding occupancies divided by two). That is, UN has a triple bond, apparently very similar to that of uranium monocarbide (UC)[55].

The low-energy spectrum of UN$^{2+}$ is characterized in the WFT calculations by states of spin-doublet multiplicity. The PT2 EBO for the ground state is 2.76 (Fig. 6b, d). This EBO is marginally smaller than that of UN even though the equilibrium bond length is slightly shorter in UN$^{2+}$ than in UN. This aspect can be understood from a closer analysis of the bonding NOs, and their populations, in Fig. 6b, d. The σ bond has a large, 40%, weight from U 5fσ (compared to 14% in the UN case), and contains only minor 6dσ character (4%) irrespective of the approach used. As already mentioned, this finding is in agreement with the comparative bond analysis based on the DFT calculations. Despite the marginally smaller numerical EBO value, the WFT calculation clearly indicates a triple bond in UN$^{2+}$ as well. The WFT and DFT calculations for the charged and neutral UN diatomic species, and the DFT calculations for the clusterfullerenes, produce consistent results concerning the U-N bonding picture. We conclude that the U-N bonds in UN@$C_s$(6)-$C_{82}$ and UN@$C_2$(5)-$C_{82}$ are genuine triple bonds.

## Discussion

The charged diatomic species UN$^{2+}$ was captured and stabilized by fullerene cages. UN@$C_s$(6)-$C_{82}$ and UN@$C_2$(5)-$C_{82}$, were synthesized and characterized by X-ray single-crystal diffraction, UV-vis-NIR spectroscopy, and Raman spectroscopy, as well as quantum-chemical calculations. The U-N bond lengths obtained for UN@$C_s$(6)-$C_{82}$ and UN@$C_2$(5)-$C_{82}$ are measured as 1.760(7) and 1.760(20) Å. These U-N bonds can be assigned as U≡N triple bonds.

The comparative study of UN@$C_s$(6)-$C_{82}$, UCN@$C_s$(6)-$C_{82}$, and U@$C_s$(6)-$C_{82}$ suggests that the interaction between the U atom and the fullerene cage is significantly weakened in the presence of the robust U≡N bond, leading to the longest metal-cage contact among all the reported uranium-based EMFs. Moreover, variable-temperature single-crystal X-ray diffraction (VT-SC-XRD) reveals a unique host-guest interaction pattern between entrapped UN$^{2+}$ and the fullerene cage: At lower temperature, the fullerene cage behaves like a ligand which

coordinates to the U≡N unit; at higher temperature, UN$^{2+}$ is more mobile and samples different uranium sites inside the fullerene cage.

Quantum-chemical calculations reveal a ground spin-doublet state for UN@$C_s$(6)-$C_{82}$ and UN@$C_2$(5)-$C_{82}$ with the unpaired electron localized at the uranium, which attains a formal +5 oxidation state (f$^1$). Calculations also show an electron transfer of 2 electrons from the UN cluster to the $C_2$(5)-$C_{82}$ and $C_s$(6)-$C_{82}$ cages, resulting in a (UN)$^{2+}$@($C_{82}$)$^{2-}$ electronic structure. The NLMO analyses for both $C_{82}$ isomers, and additional wavefunction calculations, demonstrate the presence of the U≡N triple bond, slightly weakened by metal-cage interactions when compared to free UN$^{2+}$.

This study provides new insights for both endohedral fullerenes and actinide compounds. UN@$C_s$(6)-$C_{82}$ and UN@$C_2$(5)-$C_{82}$ show that diatomic species UN$^{2+}$ can be captured and stabilized inside carbon cages. Moreover, it presents a new type of actinide compounds with a U≡N bond. The unique host-guest molecular structures revealed in this study demonstrates the exceptional ability of fullerene cages to capture metastable actinide molecules and rare bonding motifs. Ongoing studies are underway to extend the paradigm to capture other currently elusive but fundamentally important actinide bonding motifs, i.e., U≡C, by fullerene cages.

## Methods

### Synthesis and Isolation of UN@$C_s$(6)-$C_{82}$ and UN@$C_2$(5)-$C_{82}$

The carbon soot containing thorium EMFs was synthesized by the direct-current arc-discharge method. The graphite rods, packed with U$_3$O$_8$ and graphite powder (molar ratio of U/C = 1:30), were annealed in a tube furnace at 1000 °C for 20 h under an N$_2$ atmosphere and then vaporized in the arcing chamber under 200 Torr He and 4 Torr N$_2$ atmosphere. The resulting soot was extracted with CS$_2$ for 12 h. The separation and purification of UN@$C_s$(6)-$C_{82}$ and UN@$C_2$(5)-$C_{82}$ were achieved by a multistage HPLC procedure (Supplementary Figs. 1, 2). Multiple HPLC columns, including Buckyprep-M (25 × 250 mm, Cosmosil, Nacalai Tesque Inc.), Buckprep-M (10 × 250 mm, Cosmosil, Nacalai Tesque, Japan), Buckprep (10 × 250 mm, Cosmosil, Nacalai Tesque, Japan) and 5PBB (10 × 250 mm, Cosmosil, Nacalai Tesque, Japan), were utilized in this procedure. Toluene was used as the mobile phase and the UV detector was adjusted to 310 nm for fullerene detection. In total 2.02 g of graphite powder and 1.58 g of U$_3$O$_8$ (molar ratio of C:U = 30:1) were packed in each rod. On average ca. 40 mg of crude fullerene mixture per rod was obtained and totally 800 carbon rods were vaporized in this work. After HPLC isolation and purification process, ca. 0.2 mg purified UN@$C_s$(6)-$C_{82}$ and UN@$C_2$(5)-$C_{82}$ were obtained. The obtained samples show a brown color in toluene and

carbon disulfide solutions, and the color in carbon disulfide is illustrated in Fig. 4a. The sample is stable in air and no decomposition was detected after 3-month storage in the air.

## Spectroscopic studies

The positive-ion mode matrix-assisted laser desorption ionization time-of-flight (MALDI-TOF) (Bruker, Germany) was employed for the mass characterization. The UV-vis-NIR spectra of the purified UN@$C_s$(6)-$C_{82}$ and UN@$C_2$(5)-$C_{82}$ were measured in $CS_2$ solution with a Cary 5000 UV-vis-NIR spectrophotometer (Agilent, USA). The Raman spectra were obtained using a Horiba Lab RAM HR Evolution Raman spectrometer using a laser at 785 nm. The micro Fourier transform infrared spectra were obtained at room temperature using a Vertex 70 spectrometer (Bruker, Germany) with a resolution of 4 cm$^{-1}$. The morphology of samples prepared Raman testing is shown in Supplementary Fig. 20.

## ESR study

Continuous-wave (CW) EPR experiments were performed on a Bruker ElexSys E580 spectrometer at the X band ($\omega$ = 9.36 GHz) with the samples dissolved in $CS_2$. The low-temperature environment was achieved by using an Oxford Instruments ESR900 and CF935 liquid helium cryostat.

## X-ray crystallographic study

The black block crystals of UN@$C_s$(6)-$C_{82}$ and UN@$C_2$(5)-$C_{82}$ were obtained by slow diffusion of the $CS_2$ solution of the corresponding metallofullerene compounds into the benzene solution of [Ni$^{II}$(OEP)]. Single-crystal X-ray datas of UN@$C_s$(6)-$C_{82}$ and UN@$C_2$(5)-$C_{82}$ were collected using synchrotron radiation (0.82641 Å) with a MX300-HE CCD detector at beamline BL17B of the Shanghai Synchrotron Radiation Facility (SSRF). The multiscan method was used for absorption correction. The structures were solved using direct methods[56] and refined on F$^2$ using full-matrix least-squares using the SHELXL2015 crystallographic software packages[57]. Hydrogen atoms were inserted at calculated positions and constrained with isotropic thermal parameters. The cif files of six crystals in this work are shown in Supplementary Data 1 and ORTEP-style illustration with probability ellipsoids shown in Supplementary Fig. 25.

## Computational details

Geometry optimizations and vibrational normal modes were carried out with the Amsterdam Density Functional (ADF, v. 2017) package[36] using Kohn-Sham (KS) density functional theory (DFT). The Perdew-Burke-Ernzerhof (PBE)[33] functional along the all-electron triple-ζ polarized (TZP) Slater-type orbital (STO) basis sets[31,32] were employed in the calculations. The scalar-relativistic (SR) zero-order regular approximation (ZORA)[30] Hamiltonian was used to treat relativistic effects. Dispersion corrections by means of 'D3'[34,35] were included in the calculations. We tested different functionals, which gave comparable results. Additional calculations were performed using Gaussian (G16) package[37], the PBE functional, and Gaussian-type orbital (GTO) basis sets as follows: 6–31 G(d,p) for C and the SDD basis sets with a matching scalar-relativistic effective core potential for U, as provided by the G16 basis set library[38]. Selected optimized systems were subjected to natural localized molecular orbital (NLMO) bonding analyses, carried out with NBO6[49]. Raman spectra were calculated with G16, the PBE functional and the same basis sets used in the optimizations. The optimizations followed by the frequency and Raman intensity calculations were performed for UN@$C_s$(6)-$C_{82}$ and UN@$C_2$(5)-$C_{82}$.

Wavefunction theory (WFT) calculations were additionally performed to confirm the bonding analysis derived from the DFT calculations. Accordingly, the low-energy electronic structure and chemical bonding in uranium mononitride (UN) and its dication (UN$^{2+}$) were investigated with spin-orbit-coupled multiconfiguration approaches.

In the first step, a set of multiconfigurational wavefunctions were calculated with complete active space (CAS) self-consistent field (SCF) theory[58]. Accurate state-energies accounting for dynamic correlation were calculated with the extended multi state (XMS) CAS perturbation theory at second-order (XMS-CASPT2)[59], with an IPEA shift of 0 a.u. and an imaginary shift of 0.20 a.u.. All atoms were treated with the atomic natural orbital relativistic core-correlated (ANO-RCC) basis sets of polarized valence triple-zeta quality (without h-functions for U). Scalar-relativistic effects were treated via the use of the second-order Douglas-Kroll-Hess (DKH2) Hamiltonian[60–63].

Concerning UN, the active space correlated 9 electrons among 11 orbitals i.e., the N-2p and the U-5f and U-7s shells, CAS(9e, 11o). For UN$^{2+}$, a CAS(7e, 11o) active space was used, comprising the N-2p, U-5f and U-6dσ atomic orbitals. Wavefunctions were calculated in separate state-averaged runs for the quartet and doublet spin-multiplicity blocks: 5 quartets and 6 doublets for UN, and 16 quartets and 7 doublets for UN$^{2+}$. Using SF multistate XMS-CASPT2 wavefunctions and energies, SOC was treated by state interaction, using the restricted active space state-interaction (RASSI) program of the OpenMolcas suite[53]. Potential energy scans were performed along the internuclear distances for both nitrides, between 1.70-1.85 Å and 1.65–1.75 Å for UN and UN$^{2+}$ respectively. All WFT calculations were performed with OpenMolcas[64].

Key results are summarized in Supplementary Tables 16, 17 for UN and UN$^{2+}$, respectively. Figure 6 displays the corresponding ground state (GS) natural orbitals (NOs) and populations. The electronic GS of UN results from the interaction of the U$^{3+}$ 5f$^2$7s$^1$ atomic configuration with the closed-shell N$^{3-}$ configuration[65–67]. This is unlike the electronic GS of the UO isoelectronic species, for instance, which derives from the more intuitive U$^{+3}$ 5f$^3$7s$^0$ metal configuration[68,69]. Resonant two-photon ionization experiments established a (5f$^2$7s$^1$) $\Omega$ = 3.5 spin-orbit GS for UN, with an equilibrium distance of 1.765 Å[65]. The bond length is quite well reproduced by our all-electron ZORA/PBE/TZP calculations (1.756 Å), as well as by previous quasirelativistic DFT calculations based on the BP local density approximation (1.743 Å)[66] and the B3LYP global hybrid functional (1.748 Å)[67]. In full agreement with the experiments, the present WFT calculations predict an $\Omega$ = 3.5 SO GS for UN at an equilibrium optimized distance of 1.761 Å (Supplementary Fig. 19), which falls only 0.004 Å short of the gas-phase measurement. For easier comparison with the DFT calculations, we used the DFT-optimized structures for further analysis.

Metal-ligand covalent bonding occurs in UN mainly between a U 5fσ/6dσ hybrid and an N 2pσ, leading to a σ bonding MO, and between U 5fπ/6dπ hybrids and N 2pπ leading to two π bonding MOs. According to the WFT calculations, the spin-free GS (SF, i.e., without treating SOC) is a spin-quartet $\Lambda$ = 5 $^4$H state derived predominantly from the [π$^4$σ$^2$]–5f(δ$^1$φ$^1$)7s$^1$ orbital configuration (92%, Supplementary Table 16). According to the PT2 calculations, UN has an effective bond order (EBO) of 2.82 (Fig. 6a, c, determined from bonding minus antibonding occupancies divided by two). That is, UN has a triple bond, apparently very similar to that of uranium monocarbide (UC)[55]. Worth noting is that the NOs of UN are virtually identical to the NOs of UC, and the EBO of 2.82 is the same too. SOC has no net influence on the chemical bonding in UN, since the $\Omega$ = 3.5 spin-orbit GS Kramers pair derives 100% from the $\Lambda$ = 5 $^4$H spin-free state. Excited spin-orbit states are separated from the GS by at least 800 cm$^{-1}$.

The low-energy spectrum of UN$^{2+}$ is characterized in the WFT calculations by states of spin-doublet multiplicity. The equilibrium bond length is 1.705 Å (Supplementary Fig. 19), similar to what was obtained by DFT and somewhat shorter than the U-N distances in the clusterfullerenes. The GS of UN$^{2+}$ is generated by the formal U$^{5+}$ 5f$^1$ configuration and identifies with a spin-doublet $\Lambda$ = 3 $^2\Phi$ state dominated by the [π$^4$σ$^2$]–5f(φ$^1$δ$^0$) orbital configuration (85%). Excited electronic states are energetically well separated from the GS. The two states contributing to the $\Omega$ = 2.5 spin-orbit GS only differ in the

population of the U 5fϕ and 5fδ orbitals, and therefore there is no net effect from SO coupling on the GS chemical bonding.

The PT2 EBO for the ground state is 2.76 (Fig. 6b, d). This EBO is a bit smaller than that of UN even though the equilibrium bond length is slightly shorter in $UN^{2+}$ than in UN. This aspect can be understood from a closer analysis of the bonding NOs, and their populations, in Fig. 6b, d. The σ bond has a large, 40%, weight from U 5fσ (compared to 14% in the UN case), and contains only minor 6dσ character (4%) irrespective of the approach used. As already mentioned, this finding is in agreement with the comparative bond analysis based on the DFT calculations. The π bonding involves relatively evenly the U 5fπ/6dπ shells, about 22/16% (compared to 26/9% in UN), again irrespective of the approach used. As such, the shorter triple bond in free $UN^{2+}$ is due to N-2p orbital overlap mainly with the radially less extended 5f orbitals of U in $UN^{2+}$, rather than mainly with the more extended U 6d orbitals in UN. Likely, the lack of Coulomb repulsion generated by the two additional unpaired electrons localized at U in the UN case also contributes to the shorter distance for $UN^{2+}$. The stronger 5f bonding in the latter species results in an increase of static correlation which manifests itself in larger occupations of the σ and π antibonding orbitals, which in turn give a somewhat reduced effective bond order compared to UN. Despite the marginally smaller numerical EBO value, the WFT calculation clearly indicates a triple bond in $UN^{*2+}$ as well. The WFT and DFT calculations for the diatomic species, and the DFT calculations for the clusterfullerenes, are therefore confirmed to produce consistent results. We conclude that the U-N bonds in $UN@C_s(6)$-$C_{82}$ and $UN@C_2(5)$-$C_{82}$ are genuine triple bonds.

## Reporting summary

Further information on research design is available in the Nature Research Reporting Summary linked to this article.

## Data availability

The X-ray crystallographic coordinates for structures reported in this study have been deposited at the Cambridge Crystallographic Data Centre (CCDC), under deposition numbers 2120710, 2120939, 2120731, 2050571, 2120708, and 2120709. These data can be obtained free of charge from The Cambridge Crystallographic Data Centre via www.ccdc.cam.ac.uk/data_request/cif. The data that support the findings of this study are available from the corresponding authors upon request. Source Data for Fig. 4 and Supplementary Figs. 1, 2, 3, 24 are provided with this paper.

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

## Acknowledgements

We thank the staffs from BL17B1 beamline of National Facility for Protein Science in Shanghai (NFPS) at Shanghai Synchrotron Radiation Facility, for assistance during data collection. We also thank professor Bing-Wu Wang and Dr. Rong Sun for their help with ESR measurement. N.C. thanks the National Science Foundation China (NSFC 91961109, 52172051), and the Natural Science Foundation of Jiangsu Province (BK20200041). Priority Academic Program Development of Jiangsu Higher Education Institutions (PAPD). J.A. acknowledges support for the theoretical component of this work from the U.S. Department of Energy, Office of Basic Energy Sciences, Heavy Element Chemistry program, under grant DE-SC0001136. We thank the Center for Computational Research (CCR)[70] at the University at Buffalo for providing computational

resources. L.E. thanks the NSF for the generous support of this work under CHE-1801317. The Robert A. Welch Foundation is also gratefully acknowledged for an endowed chair to L.E. (grant AH-0033). DCS gives acknowledgment to the infrastructure support provided through the RECENT AIR grant agreement MySMIS no. 127324.

## Author contributions

N.C. conceived and designed the experiments. Q.Y.M., X.Y.L. and W.Y. synthesized and isolated all the compounds. J.A., L.A. and D.C.S. performed the computations and theoretical analyses. Q.Y.M. and J.X.Z. performed the single-crystal measurements. Y.R.Y. and Q.Y.M. performed the crystallographic analysis. Q.Y.M. performed the spectroscopic measurements. N.C., J.A., Q.Y.M., L.A., Y.R.Y., D.C.S., W.Y., and L.E. co-wrote the manuscript.

## Competing interests

The authors declare no competing interests.
