## [Peer Review File - updated · Nature Communications]

A charged diatomic triple-bonded $U\equiv N$ species trapped in C_{82} fullerene cagesEDITORIAL NOTE: Parts of this peer review file have been redacted at the author's request.

REVIEWER COMMENTS

Reviewer #1 (Remarks to the Author):

The authors report on the synthesis of two UN@C82 isomers. The compounds are characterized by X-ray diffraction, optical, Raman, and quantum chemical calculations. The techniques have been done competently. The computational analysis is excellent, and most of the experimental analysis is well done, except a fairly important point below. References are appropriate.

The authors have demonstrated some considerable synthetic skill to make these systems. But in the end from grams of starting materials and 800 carbon rods they isolate 0.2 mg of the title complexes (each or total I was not entirely clear on). Isolating UN inside these fullerenes is neat, but it has to be acknowledged that the incredibly low yields and scale, which limits the number of techniques that can be brought to bear, very much degrades the impact of this work, especially when isolable molecular analogs have been made on gram scales.

So it is then a reasonable question to ask what the novelty is given it is arguably easier to make regular coordination complexes since they can be made on gram scales and examined fully with a wider range of techniques than here. The authors make a variety of claims or novelty statements, but not many stand up to scrutiny, in particular the shortest UN distance claim is factually incorrect (see below). As I read through the computational analysis whilst I thought that it was a professional write-up I did not come across a new electronic structure for UN. The orbitals all look to be arranged very much as has been shown elsewhere before, comparison to which seems to be missing from the UN bonding discussion - sure some %s shift around but it all looks very similar to me.

This is a really nice and mostly solid study in many regards, but I was left with the impression that using a synthetically arduous approach the authors made a system that looks largely the same as already reported UN bonds and were unable to characterize it as much as in other studies because of the sub-mg yields. If there were no regular coordination complexes of UN then I would say this would be an advance over inert gas matrices, but that is not the case, so I do not see the novelty or new conceptual insight that I'd expect to see in a Nature family journal. I therefore have to conclude that I am unable to support publication in Nature Communications.

Specific issues to address before publication anywhere:

The first sentence of the abstract is ambiguous. It can be read as the authors saying that all actinide multiple bonds are elusive and only found in gas matrices, which is clearly not the case.

The second sentence of the abstract is worded poorly as it says that UN is unprecedented, but it would seem that Andrews has isolated it in gas matrices - the authors then go on to contradict themselves (line 69) by acknowledging the work of Andrews.

The penultimate sentence of the abstract is not worded precisely enough so could be debated. There are around 20 UN triple bonds now, is that still rare?

The last sentence is imprecise, since it implies that a UN triple bond is still elusive, but they have been made. I don't think this is what the authors meant but it is how it reads.

Introduction, Cummins made U2C2, there are now many UNU, and Ephritkhine made UCN, so how can the authors refer to those linkages as unprecedented?

Line 65, TrenTIPSUN should be mentioned overtly here.

The authors use the phrase cluster a lot, the phrase 'diatomic cluster' is a contradiction in terms and is an inappropriate description.

Lines 133-152, the authors present crystallographic bond metrics without standard uncertainties, which makes the values and statements associated with them meaningless, and in one case certainly factually incorrect. A bond distance is only fair game to refer to as shorter or longer than another bond distance if they do not overlap by the 3sigma rule. The UN distance they quote is 1.760 angstroms, undoubtedly short, however the UN distance of Mazzanti's complex is 1.769 angstroms, but her bond length has a standard uncertainty of 2, so her UN distance could be as small as 1.763 angstroms. In the 100 K cif file the UN distance is given as 1.760(9) angstroms, so the UN distance could be as long as 1.787 angstroms (it could also be shorter but that is not the point here), which also falls within the lower range of Liddle's uranium +6 nitride (1.778-1.820 angstroms by 3sigma). So the author's and Mazzanti's UN distances are statistically indistinguishable, therefore the authors cannot lay claim to have the shortest UN bond characterized by crystallography.

Did the authors observe any absorptions characteristic of the uranium +5 oxidation state in the nIR region? Whilst I am mostly convinced that they have made UN@C82 I don't think the possibility of UO@C82 being present in variable proportions can be considered to be completely excluded given the extensive crystallographic disorder and [understandable] absence of magnetic data. The optical spectrum is really the only way of the data provided to probe if it is all uranium +5 or whether there is any +4 present but this key aspect is not covered in the discussion. This would not affect the computational analysis, but purity really is an issue when only making 0.2 mg.

Line 273, as written I found this confusing. The authors say the two optical spectra of the two UN fullerene isomers are different then seem to say this shows their electronic structures are the same. If the spectra are different their electronic structures are not the same.

Raman, what do the spectra look like above 600 cm⁻¹? The authors found the UN wagging mode but what about the UN stretch? I would expect this aspect to be probed and discussed as a bare minimum.

Line 400, delete fully, that is subjective and these compounds aren't fully characterized when thinking about the range of techniques available.

Lines 405-407, the mistake of claiming the shortest UN distance is repeated, so then saying it is notably shorter is also incorrect.

Reviewer #2 (Remarks to the Author):

The manuscript is very well written and easy to read. The Experimental part is good and state-of-the-art. I am no expert for quantum-chemical calculations but they look reasonable to me, especially the agreement of the observed and simulated Raman band positions is nice.

In my opinion the manuscript is unclear when it comes to the differentiation of the encaged species, if it is "UN", "UO", or "UC" as during the synthesis also a lot of O and C atoms have been present. Maybe the authors could explicitly state if/that the mass spectroscopic analysis is evidence enough to exclude UO and UC species?

For the mass specs I miss simulations, especially for the isotopic distributions (¹³C, ...) and if the resolution was high enough to discriminate the species. With some many C atoms the ¹³C signal should superpose with the one from ¹²C, a high-res MS should however be able to discriminate the species. The authors should comment on this.

Regarding the crystal structures: That the N atom is completely ordered and the heavy U atom not is peculiar? Is that really plausible? Are there proven examples for such a observation?

Is there no O/C/N mixed site occupancy possible?

Are the electron densities of the less occupied U atom positions similar to N/O/C electron density?

The absolute densities should be given in the SI.

The authors should investigate and comment, if structure models in subgroup C2 give better results, i.e. resolution of the disorder by twinning. That would be something reasonable to expect for the

models in C2/m and should be commented.

All bond lengths and atom distances in the manuscript and the SI should be reported with their respective standard uncertainties as only these will allow for a proper comparison with other species, especially when it comes to the discussion of the UN bond lengths and its comparison with compounds from the literature.

The U-N bond length is a value that is surely biased by the disorder and should only be used with the necessary care.

How far is the residual electron density away from the U atom? Can it be clearly discriminated from a N atom?

Where would one expect the UN stretch vibration? These should be given. Can an isotope effect be observed?

The observed and calculated Raman bands should be assigned more clearly in the manuscript as well as in the SI as the arrows in Supp. Fig. 22 and 23 are simply too small.

Discrimination from UO and UC species: Where would those show bands? Have the respective UO and UC species been calculated?

Overall, I recommend revision of the manuscript. I think that if the issues can be sorted out it would be a fine contribution.

Reviewer #3 (Remarks to the Author):

In this article, Chen presents a UN triple bond captured inside fullerene cages. The isolation of uranium nitride species is of great importance because this material has been proposed as possible uranium-based nuclear fuel. Understanding the fundamentals of its bonding is thus rather useful. The results presented in this article are interesting but I disagree with most comments on the fact that they are unique opportunities to study the UN bond. Coordination compounds are now known and accessible on a larger scale, which allowed various characterization and computations. This article is following with a new type of UN terminal bond since I would argue that the U coordination to the fullerene mimics well what is observed with a classical coordination compound. If most characterizations are fair (see below), the technique only provides crystals and the compounds remain rather elusive. Considering that molecular compounds with similar bonding exist, I would state that this study merits publication in a more specialized journal provided that my few technical comments are sorted out.

1. The crystallographic disorder is large and the esd on the distance must be large as well. I have seen no mention of esd in the main text or SI. I would argue that Table S14 and the R1/wR2 give an idea of the large esd that should be associated with these distances. Without these esd comparing distances is pointless, especially if the authors want to convince us that the UN bond of this work is the shortest one reported to date! On this matter, I would be very cautious since the molecular UN bond distances are very close to that one if not identical.

2. Both crystallographic positions of the U are true. The fact that there is a minor one does not mean the distance argument can only be made from the major one. An average would be more accurate for the discussion.

3. I would agree with the authors with the hypothesis stating that UN species are formed during the arc discharging process but not that it can be related to similar distances observed in unstable gas molecules, this is too speculative.

4. The strong interaction of U with the C-skeleton of the fullerene is shown by the calculated spin doublet ground-state being the most stable one. Would it be possible to confirm this by EPR spectroscopy? Since the EPR is a sensitive technique, few crystals should be enough.
5. The VT-XRD studies are nice and interesting but do not bring much to the bonding analysis.
6. The UV-Vis-NIR data is provided without informing the reader of the nature of the bands reported.
7. I wonder if the use of the term somewhat twice to define the UN bonding nature is not too much. I would say, the bond is polarized. How does it compare with molecular coordination compounds?
8. The comparison of the NU and UN₂- electronic structure by ab initio methods is also interesting but the conclusion of this interesting comparison full of information is elaborated, neither in the conclusion nor in the discussion. Were these computations done only to mention that WFT and DFT produce similar results? Could these be gathered and compared on a table?

Reviewer #4 (Remarks to the Author):

This is another paper in the series of the synthesis of clusterfullerenes containing simple, and now characterizable, f-block diatomics with multiple metal-element bonding. This time U≡N in two different fullerene symmetries.

The driver for the work is understanding covalent bonding with the 5f and 6d orbitals in An-ligand multiple bonds. The authors are able to slow the rotation of the diatom at low temperatures, enabling them to achieve this aim, since they can quantify the contributions of the C cage.

The very short 1.760 Å UN bond is the same as the UO bond in the simplest U(VI) uranyl complexes, yet this is U(V). I think this comparison is worthy of comment. In addition to this, there is a list of points to address below. The most important being to present fully the characterizing data for the compounds.

I think this is a really elegant piece of work and look forward to reading the finished article.

Other, more detailed points to address:

1. Line 42- describe more carefully stable with respect to what.
2. Line 104. Describe the physical nature of the products. Yield. Colour (I assume they're black or very dark brown). Air-sensitivity? Include more characterizing data here, especially the Raman spec.. The synthetic description in the SI should contain pictures of the Raman spectra and as full synthetic details as the authors can manage.
3. Line 106. Please explain the reason for co-crystallization with Ni(OEP). I would also like to see use of spectroscopic characterizing data (Raman spec.?) that confirms the electronic structure of the fullerene complex is not disturbed by the presence of the Ni complex. It is known from magnetic studies that the magnetic field of paramagnetic f-block complexes can easily extend 20 Å through space. Maybe the Ni complex is actually useful in isolating the U spins from each other to give better data? Some of this should/could also go in the SI.
4. Line 133. All the bonds discussed in the paragraph should have esds.
5. Line 152. I don't understand the link between a bond length in an isolated molecule and the mechanism by which it was made.
6. Line 212. The contributions from the cage are clearly very important in stabilising, ligating, the U; this should be emphasized in the abstract. Here also, draw comparisons on U-C distances with literature organometallics U-C5 and U-C6 systems, ideally with U in the same formal oxidation state for comparison. For example U(IV) Cp₄ <https://www.degruyter.com/document/doi/10.1515/znb-1962-0410/html> and U(IV) arene in <https://pubs.acs.org/doi/abs/10.1021/acs.inorgchem.1c03365>.
7. Line 238. I'm not sure the retention of order of the NiOEP is interesting.
8. Line 242. Is it helpful to describe the N atom as stationary and U as disordered? The description on

252 is more useful, as there is still significant order in the U's position, and the cage bonding

9. Line 293. Please define the ratio of d:f in this orbital. How does it compare to the other U-E multiple bonds?

10. Line 313 5f rather than 6d backdonation? Again, please compare the size of the U contributions (4-10 %) to that found in other uranium organometallics such as the Cp and COT sandwiches.

11. Line 354 – this is the first time that UC is described as having a genuine triple bond, which I agree with, but which also undermines the statements made in the introduction about the absence of U-E triple bonds in the literature.

12. Line 417 please replace 'spins' with a more precise description, including the axis.

Response to reviewer's comments:

Reviewer #1 (Remarks to the Author):

The authors report on the synthesis of two UN@C₈₂ isomers. The compounds are characterized by X-ray diffraction, optical, Raman, and quantum chemical calculations. The techniques have been done competently. The computational analysis is excellent, and most of the experimental analysis is well done, except a fairly important point below. References are appropriate.

The authors have demonstrated some considerable synthetic skill to make these systems. But in the end from grams of starting materials and 800 carbon rods they isolate 0.2 mg of the title complexes (each or total I was not entirely clear on). Isolating UN inside these fullerenes is neat, but it has to be acknowledged that the incredibly low yields and scale, which limits the number of techniques that can be brought to bear, very much degrades the impact of this work, especially when isolable molecular analogs have been made on gram scales.

So it is then a reasonable question to ask what the novelty is given it is arguably easier to make regular coordination complexes since they can be made on gram scales and examined fully with a wider range of techniques than here. The authors make a variety of claims or novelty statements, but not many stand up to scrutiny, in particular the shortest UN distance claim is factually incorrect (see below). As I read through the computational analysis whilst I thought that it was a professional write-up I did not come across a new electronic structure for UN. The orbitals all look to be arranged very much as has been shown elsewhere before, comparison to which seems to be missing from the UN bonding discussion - sure some %s shift around but it all looks very similar to me.

This is a really nice and mostly solid study in many regards, but I was left with the

impression that using a synthetically arduous approach the authors made a system that looks largely the same as already reported UN bonds and were unable to characterize it as much as in other studies because of the sub-mg yields. If there were no regular coordination complexes of UN then I would say this would be an advance over inert gas matrices, but that is not the case, so I do not see the novelty or new conceptual insight that I'd expect to see in a Nature family journal. I therefore have to conclude that I am unable to support publication in Nature Communications.

Specific issues to address before publication anywhere:

1. The first sentence of the abstract is ambiguous. It can be read as the authors saying that all actinide multiple bonds are elusive and only found in gas matrices, which is clearly not the case.

Response: We thank the reviewer for the comments. In the first sentence, we in fact intend to imply that 'actinide diatomic molecules' rather than 'actinide multiple bonds' have only been found in the gas matrices. The original writing caused some confusion for the reviewer. Accordingly, we revised the first sentence as follows: '**Actinide diatomic molecules are ideal models to study elusive actinide multiple bonds, but most of these diatomic molecules have only been studied in solid inert gas matrices to date.**' We hope this revised sentence can convey our point without confusion.

2. The second sentence of the abstract is worded poorly as it says that UN is unprecedented, but it would seem that Andrews has isolated it in gas matrices - the authors then go on to contradict themselves (line 69) by acknowledging the work of Andrews.

Response: We have deleted the word 'unprecedented'.

3. The penultimate sentence of the abstract is not worded precisely enough so could be debated. There are around 20 UN triple bonds now, is that still rare?

Response: We have deleted the word 'rare'.

4. The last sentence is imprecise, since it implies that a UN triple bond is still elusive, but they have been made. I don't think this is what the authors meant but it is how it reads.

Response: Here, we made this statement mainly referring to the other elusive and yet to be found multiple bonded species, i.e., $U\equiv C$. We thought this method could be utilized to stabilize them in the molecular compound. To address the reviewer's concern and make it clearer, we replaced this sentence with '**Ongoing studies are underway to extend the paradigm to capture other currently elusive but fundamentally important actinide bonding motifs, i.e., $U\equiv C$, by the fullerene cages.**'

5. Introduction, Cummins made U_2C_2 , there are now many UNU, and Ephritkhine made UCN, so how can the authors refer to those linkages as unprecedented?

Response: The U_2C_2 and UCN in endohedral fullerenes, i.e., $U_2C_2@C_{78}$, $U_2C_2@C_{80}$ and $UCN@C_{82}$ have unique bonding structures. U_2C_2 in $U_2C_2@C_{78}$ and $U_2C_2@C_{80}$ demonstrates a unique bonding motif with two U bridged by a $C\equiv C$ triple bond (*J. Am. Chem. Soc.*, **141**, 20249–20260 (2019)), which, to the best of our knowledge, has not been reported by Cummins (*Dalton Trans.*, **39**, 6632-6634 (2010)). UCN in $UCN@C_{82}$ features a triangular cluster configuration with η^2 (side-on) coordination of U by a cyanide (*J. Am. Chem. Soc.*, **143**, 16226–16234 (2021)), which is different from those reported by Ephrikhine (*Dalton Trans.*, **44**, 7727-7742 (2015)). For UNU in C_{80} , we agree that the bonding structures are not substantially different from the UNU reported before and should not be mentioned here. To make this introduction more accurate, we modified the corresponding text as follows: '**The encapsulated U_2C_2 , which presents two U bridged by $C\equiv C$ triple bond, and triangular UCN cluster, which features η^2 (side-on) coordination of U by a cyanide, show novel bonding motifs for U, broadening our understanding of the bonding properties of the actinide elements.**' For details please see the first paragraph of **Introduction**.

6. Line 65, TrenTIPSUN should be mentioned overtly here.

Response: We thank the reviewer for the kind suggestion. In the revised manuscript, in

Line 65, we have added a description of [UN(Tren^{TIPS})] (the second paragraph of **Introduction**).

7. The authors use the phrase cluster a lot, the phrase ‘diatomic cluster’ is a contradiction in terms and is an inappropriate description.

Response: In endohedral fullerene studies, these encapsulated species are commonly referred to as ‘clusters’, and clusterfullerenes specifically refer to fullerenes encapsulating species formed by two or more atoms. For details please see references (*Chem. Rev.* **113**, 5989-6113 (2013)). Here, to avoid disputes, we replaced ‘diatomic cluster’ with ‘diatomic species’ in the corresponding text.

8. Lines 133-152, the authors present crystallographic bond metrics without standard uncertainties, which makes the values and statements associated with them meaningless, and in one case certainly factually incorrect. A bond distance is only fair game to refer to as shorter or longer than another bond distance if they do not overlap by the 3sigma rule. The UN distance they quote is 1.760 angstroms, undoubtedly short, however the UN distance of Mazzanti’s complex is 1.769 angstroms, but her bond length has a standard uncertainty of 2, so her UN distance could be as small as 1.763 angstroms. In the 100 K cif file the UN distance is given as 1.760(9) angstroms, so the UN distance could be as long as 1.787 angstroms (it could also be shorter but that is not the point here), which also falls within the lower range of Liddle’s uranium +6 nitride (1.778-1.820 angstroms by 3sigma). So the author’s and Mazzanti’s UN distances are statistically indistinguishable, therefore the authors cannot lay claim to have the shortest UN bond characterized by crystallography.

Response: Many thanks to the reviewers for their corrections. In the revised manuscript, we have added standard uncertainties to the bond length and distances involved in the work. Considering standard uncertainties, the UN bond in UN@C₈₂, as the reviewer pointed out, is still very short. However, compared to the UN bond in Mazzanti’s complex, it is indeed true that we cannot determine that the UN bond in UN@C₈₂ is the shortest. Thus, to avoid misstatement, we revised the corresponding text to

‘Crystallographic analysis reveals very short U-N bond lengths of 1.760(7) and 1.760(20) Å in UN@C_s(6)-C₈₂ and UN@C₂(5)-C₈₂.’ in the abstract and revised/deleted the corresponding discussion in the main text to avoid the statement of ‘the shortest’.

9. Did the authors observe any absorptions characteristic of the uranium +5 oxidation state in the nIR region? Whilst I am mostly convinced that they have made UN@C₈₂ I don’t think the possibility of UO@C₈₂ being present in variable proportions can be considered to be completely excluded given the extensive crystallographic disorder and [understandable] absence of magnetic data. The optical spectrum is really the only way of the data provided to probe if it is all uranium +5 or whether there is any +4 present but this key aspect is not covered in the discussion. This would not affect the computational analysis, but purity really is an issue when only making 0.2 mg.

Response: We have checked the literature and found that U(V) compounds usually have characteristic absorption bands at approximately 6700, 10000, and 12000 cm⁻¹ (*Chem. Rev.* **69**, 657-671 (1969); *Inorg. Chem.* **44**, 6211-6218 (2005); *Nat. Commun.* **12**, 4832 (2021)). As shown in **Response Fig. 1**, we changed the horizontal coordinate of the UV–vis–NIR absorption spectrum to the wavenumber and could not find the corresponding characteristic peaks for U(V) compounds in the near-IR region of UN@C₈₂. In fact, in the study of endohedral fullerenes, the characteristic absorption of the encaged metal ion is hardly visible in the absorption spectra because the absorption spectra of endohedral fullerenes are generally dominated by the $\pi \rightarrow \pi^*$ excitation of their carbon cage π system, and the absorption of metals mostly overlap and become invisible. (*Chem. Rev.* **113**, 5989-6113 (2013)).

The UV–vis–NIR spectra in this work also assisted us in determining the cage structure of the two isomers and the electron transfer between the clusters and the carbon cage. Although we did not observe the absorptions characteristic of the uranium +5 oxidation state, by comparing the UV–vis–NIR spectra of UN@C₈₂ with the previously reported TbCN@C₈₂ (*J. Am. Chem. Soc.* **138**, 14764–14771(2016)), which is almost identical to those of UN@C₈₂, we can deduce that the electron transfer between the cluster and carbon cages is the same in UN@C₈₂ and TbCN@C₈₂, i.e., both

are two-electron transfers, which also helps us determine that the formal oxidation state of U in UN@C₈₂ is +5.

Response Fig. 1. (a) UV-vis-NIR spectra of UN@C₂(5)-C₈₂ (left) and UN@C_s(6)-C₈₂ (right) dissolved in CS₂.

On the other hand, the exclusion of the presence of UO@C₈₂ in the sample was determined by high-resolution matrix-assisted laser desorption/ionization time-of-flight (MALDI-TOF) mass spectrometry. **Response Fig. 2** shows the theoretically calculated isotopic distributions of UN@C₈₂ and UO@C₈₂.

Response Fig. 2. (a) Isotopic distribution of UN@C₈₂ from theoretical simulations. (b) Isotopic distribution of UO@C₈₂ from theoretical simulations.

In the separation processes of UN@C_s(6)-C₈₂ (**Supplementary Fig. 1**), the sixth stage of HPLC separation was performed by recycling on a Buckyprep column. In this HPLC profile of this stage, the fraction that contains UN@C_s(6)-C₈₂ is labelled in green (**Response Fig. 3a**). The mass spectra obtained from this fraction in front of the UN@C_s(6)-C₈₂ fraction are presented in **Response Fig. 3b**. In this mass spectrum, in

addition to the mass signal of UN@C_s(6)-C₈₂ and other fullerenes, the isotopic distribution assigned to UO@C₈₂ (1237.951) can be clearly seen. This result suggests that UO@C₈₂ was also generated during the arcing process along with UN@C₈₂. However, after further purification processes, the mass spectrum of the final purified sample (**Supplementary Fig. 1-2**) only shows the isotopic distribution of UN@C₈₂, and the peak of UO@C₈₂ (1237.951) is absent, which indicates that UO@C₈₂ has been successfully removed by recycling HPLC operations.

Response Fig. 3. (a) Partial magnification of the sixth step of chromatographic separation of UN@C_s(6)-C₈₂. The fractions marked in green are UN@C_s(6)-C₈₂. (b) Mass spectra of the previous fraction of UN@C_s(6)-C₈₂ (marked by the yellow box). The inset is an enlarged view of the mass spectra at the position marked by the blue box. The complete chromatographic separation is shown in **Supplementary Fig. 1**.

10. Line 273, as written I found this confusing. The authors say the two optical spectra of the two UN fullerene isomers are different then seem to say this shows their electronic structures are the same. If the spectra are different, their electronic structures are not the same.

Response: We thank the reviewers for pointing out the ambiguity in our presentation with respect to the absorption spectroscopic analysis. In fact, the two UN@C₈₂ isomers have different spectra because of their different isomeric cage structures. On the other

hand, both isomers have the same two-electron cluster-to-cage charge transfer. Accordingly, we revised ‘UN@C₂(5)-C₈₂ shows a different absorption pattern with two well-defined peaks at 772 and 1050 nm, resembling that of TbCN@C₂(5)-C₈₂. This indicates similar isomeric and electronic structures, respectively, which are consistent with crystallographic assignments of their molecular structures and the computational results for [UN]²⁺@C₈₂²⁻.’ as **‘On the other hand, UN@C₂(5)-C₈₂ shows a different absorption pattern from UN@C_s(6)-C₈₂, with two well-defined peaks at 772 and 1050 nm, but resembles that of TbCN@C₂(5)-C₈₂. This indicates similar isomeric structures and electronic transfer between UN@C₂(5)-C₈₂ and TbCN@C₂(5)-C₈₂ and between UN@C_s(6)-C₈₂ and TbCN@C_s(6)-C₈₂. These results are consistent with the computational results for [UN]²⁺@C₈₂²⁻ (both TbCN@C₂(5)-C₈₂ and TbCN@C_s(6)-C₈₂ have two electron cluster-to-cage electron transfer) and the crystallographic assignments of their different isomeric structures of C₂(5)-C₈₂ and C_s(6)-C₈₂.’** in the **Spectroscopic Characterization** of the revised manuscript.

11. Raman, what do the spectra look like above 600 cm⁻¹? The authors found the UN wagging mode but what about the UN stretch? I would expect this aspect to be probed and discussed as a bare minimum.

Response: We thank the reviewer for the kind suggestion. Following this suggestion, we obtained the FTIR spectrum of one of the two UN@C₈₂ isomers, UN@C₂(5)-C₈₂, and the experimental and simulated spectra are shown below in **Response Fig. 4**. The peak at 924 cm⁻¹ can be assigned to the stretching vibration of U≡N, which is close to the vibration of the U≡N triple bond in the FTIR spectrum of [UN(Tren^{TIPS})] [Na(12C₄)₂] at 936 cm⁻¹ (*Science* **337**, 717-720 (2012)). The spectrum above 1000 cm⁻¹ corresponds to the vibrations of the carbon cage, reproduced well by the theoretical calculation. We added the corresponding discussion of FTIR spectra and figures after the Raman section in the revised manuscript (**Spectroscopic Characterizations** and **Supplementary Fig. 25**).

Response Fig. 4. (a) Observed and theoretically predicted vibrational features of UN@C₂(5)-C₈₂. The lower curve (black) presents the observed infrared absorption (IR) spectrum vs. wavenumber from 1600 to 600 cm⁻¹, with quantum-chemical density-functional simulation upper (in blue).

12. Line 400, delete fully, that is subjective and these compounds aren't fully characterized when thinking about the range of techniques available.

Response: We have deleted this sentence in the revised manuscript.

13. Lines 405-407, the mistake of claiming the shortest UN distance is repeated, so then saying it is notably shorter is also incorrect.

Response: We deleted 'The bond length of 1.760 Å is, to the best of our knowledge, the shortest U-N bond length, characterized by crystallography, notably shorter than those discovered in coordination compounds but comparable to the calculated bond lengths for gas-phase molecules previously studied by matrix isolation.' in the revised manuscript.

Reviewer #2 (Remarks to the Author):

The manuscript is very well written and easy to read. The Experimental part is good and state-of-the-art. I am no expert for quantum-chemical calculations but they look reasonable to me, especially the agreement of the observed and simulated Raman band

positions is nice.

1. In my opinion the manuscript is unclear when it comes to the differentiation of the encaged species, if it is "UN", "UO", or "UC" as during the synthesis also a lot of O and C atoms have been present.

Maybe the authors could explicitly state if/that the mass spectroscopic analysis is evidence enough to exclude UO and UC species?

Response: We did only give a brief discussion about the purification process and mass spectra of UN@C₈₂, which did not elaborate them in detail. In fact, high-resolution mass spectra can unambiguously differentiate UN@C₈₂, UO@C₈₂ and UC@C₈₂. UO@C₈₂ was generated during the arcing process and was removed during the purification process. However, there is no evidence that UC@C₈₂ was generated during the arcing process in this work.

Response Fig. 5 shows the theoretically calculated isotopic distributions of UC@C₈₂, UN@C₈₂ and UO@C₈₂, which are distinguishable in the high-resolution mass spectra.

Response Fig. 5. (a) Isotopic distribution of UC@C₈₂ from theoretical simulations. (b) Isotopic distribution of UN@C₈₂ from theoretical simulations. (c) Isotopic distribution of UO@C₈₂ from theoretical simulations.

For UC@C₈₂, the mass spectral signal of 1234.042 was not observed throughout the entire HPLC separation process.

For UO@C₈₂, in the separation of UN@C_s(6)-C₈₂ (**Supplementary Fig. 1**), the sixth HPLC stage, which was performed on a Buckyprep column, the mass signal of UO@C₈₂ can be observed in fraction 6-1, while UN@C_s(6)-C₈₂ was found mainly in

fraction 6-2 labelled in green. As shown in **Response Fig. 6**, the isotopic distribution of 1237.951 can be assigned to UO@C_{82} , which overlaps with a minor isotopic distribution of 1235.962, mass signal of $\text{UN@C}_s(6)\text{-C}_{82}$. In addition, no mass signal of UO@C_{82} was observed in any of the HPLC fractions after this separation process of $\text{UN@C}_2(5)\text{-C}_{82}$ (**Supplementary Fig. 2**), which indicates that UO@C_{82} was removed in the fourth step (Buckyprep column recycling).

Response Fig. 6. (a) The HPLC profile of the sixth step of the $\text{UN@C}_s(6)\text{-C}_{82}$ chromatographic separation. The fractions marked in green mainly contain $\text{UN@C}_s(6)\text{-C}_{82}$. (b) Mass spectra of fraction 6-2. The inset is an enlarged view of the mass spectra at the position marked by the blue square. The complete chromatographic separation HPLC profiles and corresponding mass spectra are shown in **Supplementary Fig. 2**.

Accordingly, we revised the ‘**Synthesis and Isolation of $\text{UN@C}_s(6)\text{-C}_{82}$ and $\text{UN@C}_2(5)\text{-C}_{82}$** ’ section and added the following sentences: ‘It is noteworthy that UO@C_{82} is also observed during the HPLC separation process, possibly due to the leak of air into the arcing chamber, but was removed during the purification processes (**Supplementary Fig. 3a-b**). The purity of the samples was confirmed by single peak HPLC chromatography. Furthermore, the high-resolution mass spectrum of the final purified sample also shows that the isotopic distribution of the samples obtained experimentally is consistent with the theoretical isotopic distribution of UN@C_{82} , excluding the existence of UC@C_{82} or UO@C_{82} (see **Supplementary Fig. 3c-d**).’

Supplementary Fig. 3c-d. HPLC chromatograms of purified UN@C₅(6)-C₈₂ and UN@C₂(5)-C₈₂. (c) UN@C₅(6)-C₈₂ on a Buckyprep column and (d) UN@C₂(5)-C₈₂ on a 5PBB column with toluene as the eluent. HPLC conditions, $\lambda = 310$ nm; flow rate, 4 mL/min. The insets show the positive-ion mode MALDI-TOF mass spectra and expansions of the corresponding experimental isotopic distributions of the compound in comparison with their calculated values.

2. For the mass specs I miss simulations, especially for the isotopic distributions (¹³C...) and if the resolution was high enough to discriminate the species. With some many C atoms the ¹³C signal should superpose with the one from ¹²C, a high-res MS should however be able to discriminate the species. The authors should comment on this.

Response: We thank the reviewer for the kind suggestion. As commented by the reviewer, in the mass spectrum of the UN@C₈₂ molecule, we observed the ¹³C signal superposed with the signal from ¹²C.

Response Fig. 7. Isotopic distribution of UN@C₈₂ from theoretical simulations and the masses

corresponding to the major isotopic peaks.

Response Fig. 7 shows the mass spectrum and data of the isotopic distribution of the theoretical calculated UN@C₈₂. In the following discussion, only the isotopic distribution of C is considered, assuming that only ²³⁸U and ¹⁴N are in the molecule, to reduce the complexity of the discussion. We can see that the first distinct peak (near 1236) corresponds to ²³⁸U¹⁴N@¹²C₈₂, and the subsequent peaks are ²³⁸U¹⁴N@¹²C₈₁¹³C (near 1237), ²³⁸U¹⁴N@¹²C₈₀¹³C₂ (near 1238)....

The abundances of ¹³C and ¹²C in nature are 98.93% and 1.07%, respectively. Then, the probability of having only ¹²C in UN@C₈₂ (²³⁸U¹⁴N@¹²C₈₂) is given as

$$P_1 = (0.9893)^{82}$$

The probability of having exactly one ¹³C atom in UN@C₈₂ (²³⁸U¹⁴N@¹²C₈₁¹³C) is therefore

$$P_2 = 82 \times (0.9893)^{81} \times 0.0107$$

and the ratio P₂/P₁ is given as

$$\frac{P_2}{P_1} = \frac{82 \times (0.9893)^{81} \times 0.0107}{(0.9893)^{82}} = 0.8869$$

If the monoisotopic peak at 1236 assigned to ²³⁸U¹⁴N@¹²C₈₂ is regarded as 100%, the 1237 peak assigned to ²³⁸U¹⁴N@¹²C₈₁¹³C will have 88.69% relative intensity. (Gross, J. H. *In Mass Spectrometry: A Textbook*; Gross, J. H., Ed.; Springer Berlin Heidelberg: Berlin, Heidelberg, 2011; pp 67-116.) The relative intensity of the peak of UN@C₈₂ at 1237 is 89%, as shown in the above figure. If we consider the effect of the isotopes of U and N, the result should be corrected to 89%. The relative intensities of the peaks of other isotopic distributions of UN@C₈₂ can be calculated according to this method as well.

Response Table 1. Experimental and theoretical isotopic distribution of UN@C₈₂ and its relative intensity.

Isotopic distribution for UN@C₈₂		
Theoretical simulation	UN@C ₂ (5)-C ₈₂	UN@C _s (6)-C ₈₂

Neutral Mass	Intensity	m/z	Intensity	m/z	Intensity
1233.045	0.73				
1234.045	0.65				
1235.045	0.28				
1236.045	100.00	1236.160	939 (100.00%)	1236.334	7936 (100.00%)
1237.045	89.00	1237.161	818 (87.11%)	1237.334	6488 (81.75%)
1238.045	39.15	1238.170	335 (35.68%)	1238.338	2759 (34.76%)
1239.045	11.34	1239.180	39 (4.15%)	1239.348	602 (7.59%)
1240.045	2.43			1240.343	78 (0.98%)
1241.045	0.41				
1242.045	0.06				

As shown in **Supplementary Fig. 1**, the isotopic distribution of UN@C₈₂ obtained by MALDI-TOF mass spectrometry is almost identical to that of the theoretical simulation. The values and relative intensities of each peak in the isotopic distributions of UN@C₂₍₅₎-C₈₂ and UN@C₅₍₆₎-C₈₂ are listed in **Response Table 1**. Thus, the purity of UN@C₈₂ can be determined by high-resolution mass spectrometry.

3. Regarding the crystal structures: That the N atom is completely ordered and the heavy U atom not is peculiar? Is that really plausible? Are there proof examples for such an observation?

Response: The problem of disordered metal sites is commonly reported in the studies of endohedral metallofullerenes. In the majority of nitride clusterfullerenes, the N atom of the nitride cluster is generally fully ordered and located in the center of the fullerene cage. In contrast, in many cases, the metal sites inside the carbon cage are disordered, which is partially caused by the thermal vibration and partially related to the interaction between the metal and the neighboring carbon-cage moiety (*Chem. Rev.* **113**, 5989-6113 (2013)).

The energy barriers of endohedral fullerenes with metal located near different carbon cage moieties are generally not high, and thus, the metal ions can overcome the

energy barriers, resulting in the motion of the metal ion and disordered metal sites in different positions inside fullerene cages, as commonly seen in EMF crystals. However, in some cases, when the metal atoms have a strong enough interaction with a particular moiety of the carbon cage, the metal atoms can remain fully ordered. For example, in the crystal of $\text{Sc}_2@C_{2v}(4059)\text{-C}_{66}$, the two Sc atoms are fully ordered and located over the pair of doubly fused-pentagon moieties of the $C_{2v}(4395)\text{-C}_{66}$ cage because the interaction between the Sc atoms and such a fused-pentagon moiety is much stronger than others (*J. Am. Chem. Soc.* 2014, **136**, 21, 7611–7614).

In this work, although the cages of $\text{UN}@C_2(5)\text{-C}_{82}$ and $\text{UN}@C_s(6)\text{-C}_{82}$ do not have fused-pentagon moieties, the major U sites show much higher occupancies than the other minor sites, suggesting that the U sites in these two cages are largely ordered and that the U ions and neighboring carbon cage moieties have relatively strong interactions. However, when temperatures increase, as we see in the VT-crystallography, U ions can overcome the energy barriers, and the U sites become disordered.

4. Is there no O/C/N mixed site occupancy possible?

Response: Indeed, it is difficult to distinguish the case of O/C/N mixed sites in the crystal. However, in this work, the possibility of the central O/C can be excluded by high-resolution mass spectrometry, and the purities of the two $\text{UN}@C_{82}$ isomers have been confirmed by the single HPLC peak, as we have answered in detail in response to question 1.

5. Are the electron densities of the less occupied U atom positions similar to the N/O/C electron density?

The absolute densities should be given in the SI.

Response: (1) All studies thus far have shown that when clusters are embedded in the carbon cage, the nonmetallic atoms are located at the center of the carbon cage. Therefore, the low-occupancy sites close to the side of the carbon cage are reasonably determined to be disordered metal sites, as the metal ion needs to be close to cage carbon to maintain the interaction between them, which is essential to the stabilization

of the endohedral fullerene molecular structure (*Chem. Rev.* **113**, 5989-6113 (2013)).

(2) We thank the reviewer for the suggestions. However, after careful analysis, we can only provide the absolute density of the U site with the lowest occupancy in the crystal of UN@C_s(6)-C₈₂, i.e., U4 (0.0566), with an absolute density of 6.21. These data were added as a footnote to Supplementary Table S6. The reason for this is because the electron density of U is much heavier than those of N/O/C, and even the U4 site in the crystal of UN@C_s(6)-C₈₂, with the lowest occupancy, is still heavier than a complete carbon atom (with an average density of ~3). Thus, we could not provide the electron densities of the other U sites because when we removed these U sites, the structural model collapsed, and the refinement process was unable to converge.

The occupancies of all the U sites in the two crystals are shown in Response Table 2 for reference.

Response Table 2. Metal site occupancy in UN@C₈₂.

UN@C ₂ (5)-C ₈₂		UN@C _s (6)-C ₈₂	
site	occupancy	site	occupancy
U1	0.312	U1	0.6442
U2	0.188	U2	0.1903
		U3	0.1087
		U4	0.0566

6. The authors should investigate and comment, if structure models in subgroup C₂ give better results, i.e., resolution of the disorder by twinning. That would be something reasonable to expect for the models in C₂/m and should be commented.

Response: As we answered in response to question 3, the disorder of encapsulated metal is partially caused by the thermal vibration and partially related to the interaction between the metal and the neighboring carbon-cage moiety, instead of being caused by twinning. The space group of C₂/m is very commonly seen in the EMF-Ni(OEP) cocrystal system (*Chem. Rev.* **113**, 5989-6113 (2013)). The removal of the mirror to subgroup C₂ does not help solve the disorder problem but instead increases the

difficulty of refining the structure. Therefore, it is more reasonable to solve and refine the crystal in the higher $C2/m$ space group, which is also the reason why the $C2/m$ space group, rather than $C2$, is much more commonly seen in the EMF-Ni(OEP) cocrystal system.

7. All bond lengths and atom distances in the manuscript and the SI should be reported with their respective standard uncertainties as only these will allow for a proper comparison with other species, especially when it comes to the discussion of the UN bond lengths and its comparison with compounds from the literature.

Response: We thank the reviewers for this suggestion. We have added the corresponding standard uncertainties to the bond lengths and atomic distances involved in this work. Please see the revised manuscript.

8. The U-N bond length is a value that is surely biased by the disorder and should only be used with the necessary care.

How far is the residual electron density away from the U atom? Can it be clearly discriminated from a N atom?

Response: We thank the reviewer for the kind suggestion. (1) As replied in point 3, in the crystal of endohedral fullerenes, the metal disorder is mainly related to the interaction between the metal and the neighboring carbon-cage moiety, especially when the crystal is measured at low temperature (*Chem. Rev.* **113**, 5989-6113 (2013)). Therefore, the metal disorder generally reflects different conformations of the molecule with small energy barriers. In this work, we focus on the major site because it represents the most stable conformation. Thus, we discuss the UN bond length with the major U site. In the process of solving the two crystals, we found that the major U sites are completely fixed and are little affected by the disordered sites. In addition to the determined U disordered sites with small atomic displacement parameters, we did not observe a large residual electron density around U and the central N, and the top ten residual electron densities are all distributed on the fullerene cage. Therefore, we think the U-N bond lengths determined by the major U sites are accurate for the most stable

conformations of these two compounds. It is also a commonly accepted method in the studies of endohedral fullerenes.

(2) In this work, the U atom can be easily discriminated from the central N atom. In response to question 3, the metal atoms are located on one side of the carbon cage, while the nonmetal atoms are normally located at the center of the carbon cage. In addition to the determined U disordered sites with small atomic displacement parameters, we did not observe large residual electron density around U and the central N, and the top ten residual electron densities are all distributed on the fullerene cage.

9. Where would one expect the UN stretch vibration? These should be given. Can an isotope effect be observed?

Response: We thank the reviewer for the kind suggestion. Following this suggestion, we obtained the FTIR spectrum of one of the two UN@C₈₂ isomers, UN@C₂(5)-C₈₂, and the experimental and simulated spectra are shown below in **Response Fig. 8**. The peak at 924 cm⁻¹ can be assigned to the stretching vibration of U≡N, which is close to the vibration of the U≡N triple bond in the FTIR spectrum of [UN(Tren^{TIPS})] [Na(12C₄)₂] at 936 cm⁻¹ (*Science* **337**, 717-720 (2012)). The spectrum above 1000 cm⁻¹ corresponds to the vibrations of the carbon cage, reproduced well by the theoretical calculation. We added the corresponding discussion of FTIR spectra and figures after the Raman section in the revised manuscript (**Spectroscopic Characterizations** and **Supplementary Fig. 25**).

Response Fig. 8. (a) Observed and theoretically predicted vibrational features of UN@C₂(5)-C₈₂.

The lower curve (black) presents the observed infrared absorption (IR) spectrum vs. wavenumber from 1600 to 600 cm^{-1} , with quantum-chemical density-functional simulation upper (in blue).

We note that in the previous report by Liddle et al., the ^{15}N isotope was used in the synthesis of $[\text{UN}(\text{Tren}^{\text{TIPS}})][\text{Na}(\text{12C4})_2]$ and $[\text{UN}(\text{Tren}^{\text{TIPS}})]$ to study the isotopic effect of the vibrational band of the U-N triple bond, in which isotopic effects cause the peaks to shift to lower wavenumbers. (*Science* **337**, 717-720 (2012); *Nat. Chem.* **12**, 962-967 (2020)). However, in this work, we cannot expect the observation of a similar isotopic effect because we used regular $^{14}\text{N}_2$ in the synthesis of UN@C_{82} . In addition, the very limited amount of sample also made it very difficult to observe such an isotopic effect.

10. The observed and calculated Raman bands should be assigned more clearly in the manuscript as well as in the SI as the arrows in Supp. Fig. 22 and 23 are simply too small.

Response: We thank the reviewer for the kind suggestion. In the revised Supplementary Information, we have replaced Supp. Fig. 22 and 23 with improved figures, which hopefully will help the reader to better understand the Raman vibration modes in UN@C_{82} .

11. Discrimination from UO and UC species: Where would those show bands? Have the respective UO and UC species been calculated?

Response: As we explained in our response to the reviewer's first question, the characterization of the fractions obtained by chromatographic separation by matrix-assisted laser desorption/ionization time-of-flight (MALDI-TOF) mass spectrometry allowed us to determine that we obtained UN@C_{82} without the mixture of UO@C_{82} and UC@C_{82} , and therefore, we did not consider these substances in our theoretical calculations.

Overall, I recommend revision of the manuscript. I think that if the issues can be sorted out it would be a fine contribution.

Reviewer #3 (Remarks to the Author):

In this article, Chen presents a UN triple bond captured inside fullerene cages. The isolation of uranium nitride species is of great importance because this material has been proposed as possible uranium-based nuclear fuel. Understanding the fundamentals of its bonding is thus rather useful. The results presented in this article are interesting but I disagree with most comments on the fact that they are unique opportunities to study the UN bond. Coordination compounds are now known and accessible on a larger scale, which allowed various characterization and computations. This article is following with a new type of UN terminal bond since I would argue that the U coordination to the fullerene mimics well what is observed with a classical coordination compound. If most characterizations are fair (see below), the technique only provides crystals, and the compounds remain rather elusive. Considering that molecular compounds with similar bonding exist, I would state that this study merits publication in a more specialized journal provided that my few technical comments are sorted out.

1. The crystallographic disorder is large and the esd on the distance must be large as well. I have seen no mention of esd in the main text or SI. I would argue that Table S14 and the R1/wR2 give an idea of the large esd that should be associated with these distances. Without these esd comparing distances is pointless, especially if the authors want to convince us that the UN bond of this work is the shortest one reported to date! On this matter, I would be very cautious since the molecular UN bond distances are very close to that one if not identical.

Response: Many thanks to the reviewers for their corrections. In the revised manuscript, we have added standard uncertainties to the bond length reported in the work. In addition, to avoid misstatement, we have changed the expression 'shortest' to more precise 'one of the shortest bond lengths' or 'very short' for the description of the UN triple bond in the corresponding text.

2. Both crystallographic positions of the U are true. The fact that there is a minor one does not mean the distance argument can only be made from the major one. An average would be more accurate for the discussion.

Response: In the crystal of endohedral fullerenes, the metal disorder is mainly related to the interaction between the metal ions and the neighboring carbon-cage moieties, especially when the crystal is measured at low temperature (*Chem. Rev.* **113**, 5989-6113 (2013)). Therefore, the metal disorder generally reflects the different conformations of the molecule with small energy barriers. In this work, we focus on the major site because it represents the most stable conformation and thus show the UN bond using the major U site. In addition, in the process of solving the two crystals, we found that the major U sites are completely fixed and are little affected by the disordered sites. In addition, in addition to the determined U disordered sites with small atomic displacement parameters, we did not observe a large residual electron density around U and the central N, and the top ten residual electron densities are all distributed on the fullerene cage. Therefore, we believe that the U-N bond lengths determined by the major U sites are accurate for the most stable conformations of these two compounds. It is also a commonly accepted method in the studies of endohedral fullerenes (*J. Am. Chem. Soc.*, **137**, 10116–10119 (2015), *J. Am. Chem. Soc.*, **138**, 13030–13037 (2016), *Nat Commun* **5**, 3568 (2014)). Nevertheless, we also add bond lengths measured with minor U sites in the SI.

3. I would agree with the authors with the hypothesis stating that UN species are formed during the arc discharging process but not that it can be related to similar distances observed in unstable gas molecules, this is too speculative.

Response: To address this concern, we deleted the corresponding text ‘The similarity of the bond lengths of encaged UN to those of isolated molecules might indicate that the isolated UN species were formed during the arc discharging process and then the nanocavity of the fullerene cage captured and stabilized them in the form of endohedral fullerenes.’ We hope in this way, we have deleted the speculative discussion.

4. The strong interaction of U with the C skeleton of the fullerene is shown by the calculated spin doublet ground state being the most stable one. Would it be possible to confirm this by EPR spectroscopy? Since the EPR is a sensitive technique, few crystals should be enough.

Response: We thank the reviewers for this suggestion. Following this suggestion, we carried out an EPR test of UN@C₈₂ at low temperature. However, this attempt to further resolve the electronic structure using EPR spectroscopy was unsuccessful, as no clearly defined signal was observed even at 4 K. A similar situation occurred when we investigated the ESR signal corresponding to U(V) in our study of U₂C@C₈₀ (*Nat. Commun.* **9**, 2753-2760). The reason for the absence of the EPR signal is yet to be understood. Nevertheless, we added the sentence ‘**Attempts to further resolve the electronic structure using EPR spectroscopy were unsuccessful, as no clearly defined signal was observed at 4 K (Supplementary Fig. 26). The ESR signal corresponding to U(V) was also not observed in U₂C@C₈₀, probably due to the shielding effect of the carbon cage.**’ in the section ‘**Spectroscopic Characterizations**’ as well as **Supplementary Fig. 26** in the Supporting Information.

Response Fig. 9. X-Band EPR spectrum of UN@C₂(5)-C₈₂. It is recorded in a toluene glass tube at 4 K and 10 K.

5. The VT-XRD studies are nice and interesting but do not bring much to the bonding analysis.

Response: It is true that the VT-XRD study did not help us to study the nature of the

UN triple bond, but variable-temperature XRD is a powerful tool to unravel the temperature-dependent dynamics of endohedral metallofullerenes in the crystal lattice. In this work, the VT-XRD study of UN@C_s(6)-C₈₂ revealed a rare phase transition process and thus gives important insight into the problem of disorder impeding metallofullerene crystallography. UN@C₈₂ is the simplest endohedral clusterfullerenes found to date, and its cluster contains only one UN triple bond. The VT-XRD study of UN@C_s(6)-C₈₂ and UN@C₂(5)-C₈₂ helped us to study the movement of the UN bonding motif within the carbon cage. The U ion appears to “rotate” around the Ni···N axis, thus making the movement of the UN cluster look like a spinning top with N atoms as the apex. Therefore, VT-XRD helped us recognize the temperature-dependent dynamics of UN@C₈₂. However, the reviewer's comment made us realize that the VT-XRD maybe too extensive. Thus, in the revised manuscript, we deleted the discussion of the phase transition, which is not related to the bonding analysis.

6. The UV–Vis–NIR data is provided without informing the reader of the nature of the bands reported?

Response: We thank the reviewer for the kind suggestion. The absorption spectra of fullerene are dominated by the $\pi \rightarrow \pi^*$ excitation of their carbon π -system. When the number of carbon atoms on the fullerene cage (e.g., C₈₀ and C₈₂) or the symmetry of the carbon cage (e.g., C_s(6)-C₈₂ and C₂(5)-C₈₂ in this work) changes, a change in the absorption spectrum can be clearly observed (*Chem. Rev.* **113**, 5989-6113 (2013)). In this work, the absorption spectra obtained from experimental tests are consistent with those previously reported for TbCN@C₈₂ [REDACTED] (*J. Am. Chem. Soc.* **138**, 14764-14771 (2016)), suggesting that the carbon cage structures of the two UN@C₈₂ isomers are C_s(6)-C₈₂ and C₂(5)-C₈₂, respectively. It can also be deduced that their electronic structures are similar to those of TbCN@C₈₂, as clusters transfer two electrons to the carbon cage: [UN]²⁺@ [C₈₂]²⁻.

Considering all this information, we did not carry out calculations on the electronic spectra. Nevertheless, following this comment, we added ‘**The absorption features of the two isomers of UN@C₈₂ are dominated by the $\pi \rightarrow \pi^*$ excitation of their carbon π -system, as commonly known for other reported endohedral fullerenes.**’ in the

first paragraph of the ‘**Spectroscopic Characterizations**’ section. We hope in this way that the readers can better understand the nature of these absorption bands.

FIGURE REDACTED AT AUTHOR'S REQUEST

7. I wonder if the use of the term somewhat twice to define the UN bonding nature is not too much. I would say, the bond is polarized. How does it compare with molecular coordination compounds?

Response: We thank the reviewer for the kind suggestion. **We revised the manuscript in lines 294 and 295 to avoid vague language.**

For bond comparisons, there are works by Liddle with ^{15}N NMR (*Nat Commun.* **12**, 5649(2021)), Hayton and some of us on the ^{15}N bonds with thorium (*Chem. Sci.*, **12**, 14383-14388(2021), *Chem. Sci.*, **10**, 6431-6436(2019)), and related systems that one of us studied with Hayton and Neidig with U-C bonds (*Inorg. Chem.* **60**, 20, 15413-15420(2021), *Inorg. Chem.* **60**, 16, 12436–12444(2021), *Chem. Eur. J.* **27**, 5885(2021), *Angew. Chem. Int. Ed.* **59**, 13586(2020), *Angew. Chem. Int. Ed.* **58**, 10266(2019)). Comparing our system to what was reported in the literature in other complexes, the UN bond appears remarkably unpolarized. We have included a comment about it in the manuscript, along with relevant references.

8. The comparison of the NU and UN^{2-} electronic structure by ab initio methods is also

interesting but the conclusion of this interesting comparison full of information is elaborated, neither in the conclusion nor in the discussion. Were these computations done only to mention that WFT and DFT produce similar results? Could these be gathered and compared on a table?

Response: The comparison of the electronic structure between UN and UN²⁻ by density functional theory (DFT) and ab initio wavefunction (WFT) was indeed performed to ensure that the results derived from the DFT calculations are reliable. We edited and clarified the text and moved most of the discussion of the WFT calculations to the Supporting Information because the additional WFT results appear to have distracted the Reviewer from the main message, which is that UN in the clusterfullerenes and in the gas phase unambiguously can be assigned a triple bond. The main results from the WFT calculations are reported in Figure 5 and the Tables that were moved to the SI, and therefore we deem it unnecessary to create an additional Table.

Reviewer #4 (Remarks to the Author):

This is another paper in the series of the synthesis of clusterfullerenes containing simple, and now characterizable, f-block diatomics with multiple metal-element bonding. This time U≡N in two different fullerene symmetries.

The driver for the work is understanding covalent bonding with the 5f and 6d orbitals in An-ligand multiple bonds. The authors are able to slow the rotation of the diatom at low temperatures, enabling them to achieve this aim, since they can quantify the contributions of the C cage.

The very short 1.760 Å UN bond is the same as the UO bond in the simplest U(VI) uranyl complexes, yet this is U(V). I think this comparison is worthy of comment. In addition to this, there is a list of points to address below. The most important being to present fully the characterizing data for the compounds.

I think this is a really elegant piece of work and look forward to reading the finished article.

Response: We thank the reviewer for this comment. In the revised manuscript, we added the comment ‘These short UN bonds are similar to the situation of the short UO bond in the simplest U(VI) uranyl complexes, such as U(VI)O₂(^tBuacnac)₂ which has a UO bond with bond length of 1.770(3) Å.

Other, more detailed points to address:

1. Line 42- describe more carefully stable with respect to what.

Response: We thank the reviewer for the kind suggestion. We have added a description of "stable" in **Line 42**. The corresponding sentence has been changed from ‘Our recent studies showed that very diverse actinide clusters containing important new actinide bonding motifs can be formed and stabilized inside the fullerene cages and can thus be systematically characterized in the form of molecular compounds.’ to ‘**Our recent studies showed that very diverse actinide clusters containing important new actinide bonding motifs can be formed and stabilized inside the fullerene cages by electron transfer between the cluster and carbon cage and by the U-fullerene coordination. They can thus be systematically characterized in the form of molecular compounds.**’

2. Line 104. Describe the physical nature of the products. Yield. Colour (I assume they’re black or very dark brown). Air sensitivity? Include more characterizing data here, especially the Raman spec. The synthetic description in the SI should contain pictures of the Raman spectra and as full synthetic details as the authors can manage.

Response: We thank the reviewer for this kind suggestion. Accordingly, we added a detailed description of the physical nature of the products in SI (page 5) as follows: In total, 2.02 g of graphite powder and 1.58 g of U₃O₈ (molar ratio of C:U = 30:1) were packed in each rod. On average, ca. 40 mg of crude fullerene mixture per rod was obtained, and 800 carbon rods were vaporized in this work. After HPLC isolation and purification, ca. Purified UN@C₅(6)-C₈₂ and UN@C₂(5)-C₈₂ (0.2 mg) were obtained. The obtained samples show a brown color in toluene and carbon disulfide solutions,

and the color in carbon disulfide is illustrated in **Supplementary Fig. 20**. The sample was stable in air, and no decomposition was detected after 3 months of storage in the air.

The Raman spectra of the samples can be seen in **Supplementary Fig. 21a-b**, and we have also added pictures (**Supplementary Fig. 21f**) of the morphology of the samples made during the Raman characterization, which we hope will help readers better understand our work.

Response Fig. 11. Shape of the sample during Raman testing.

3. Line 106. Please explain the reason for co-crystallization with Ni(OEP). I would also like to see use of spectroscopic characterizing data (Raman spec.?) that confirms the electronic structure of the fullerene complex is not disturbed by the presence of the Ni complex. It is known from magnetic studies that the magnetic field of paramagnetic f-block complexes can easily extend 20 Å through space. Maybe the Ni complex is actually useful in isolating the U spins from each other to give better data? Some of this should/could also go in the SI.

Response: We thank the reviewer for the question. The reason for using Ni(OEP) for co-crystallization is to prevent the rotation of the fullerene cage, which has long been a great challenge for solving the crystal structures of these compounds. Ni(OEP), which has a noncovalent π - π interaction with endohedral fullerenes when forming a cocrystal,

was utilized in the crystal growth process to hinder the rotation of the fullerene cage.

Except for the single crystal characterization where Ni(OEP) is used as a eutectic agent, the rest of the characterizations in this work are all performed on the pristine UN@C₈₂ molecule. Thus, there is no influence of Ni(OEP) on the determination of their electronic structures. In the revised manuscript, we also carried out an EPR test of UN@C₈₂ at low temperature. However, this attempt to further resolve the electronic structure using EPR spectroscopy was unsuccessful, as no clearly defined signal was observed even at 4 K. A similar situation occurred when we investigated the ESR signal corresponding to U(V) in our study of U₂C@C₈₀ (*Nat. Commun.* **9**, 2753-2760). The reason for the absence of the EPR signal is yet to be understood. Nevertheless, we added the sentence ‘Attempts to further resolve the electronic structure using EPR spectroscopy were unsuccessful, as no clearly defined signal was observed at 4 K (Supplementary Fig. 26). The ESR signal corresponding to U(V) was also not observed in U₂C@C₈₀, probably due to the shielding effect of the carbon cage.’ in the section ‘Spectroscopic Characterizations’ as well as Supplementary Fig. 26 in the Supporting Information.

Response Fig. 12. X-Band EPR spectrum of UN@C₂(5)-C₈₂. It is recorded in a toluene glass tube at 4 K and 10 K.

4. Line 133. All the bonds discussed in the paragraph should have esds.

Response: We thank the reviewers for this suggestion. We have added the corresponding standard uncertainties to all the bond lengths reported in this work in the

revised manuscript.

5. Line 152. I don't understand the link between a bond length in an isolated molecule and the mechanism by which it was made.

Response: We thank the reviewer for the comment. Reviewer 3 raised similar questions and pointed out that this link seems to be too speculative. Thus, to make a fact-based statement and remove the speculative mechanism, we deleted the speculative discussion about this link.

6. Line 212. The contributions from the cage are clearly very important in stabilising, ligating, the U; this should be emphasized in the abstract. Here also, draw comparisons on U-C distances with literature organometallics U-C5 and U-C6 systems, ideally with U in the same formal oxidation state for comparison. For example, U(IV) Cp4 <https://www.degruyter.com/document/doi/10.1515/znb-1962-0410/html> and U(IV) arene in <https://pubs.acs.org/doi/abs/10.1021/acs.inorgchem.1c03365>.

Response: We thank the reviewer for the kind suggestion. (1) Following this suggestion, we revised the second sentence of the abstract as follows: 'Herein, we report that the U \equiv N diatomic species captured in two different fullerene cages and stabilized by the U-fullerene coordination.'

(2) Following the reviewer's comments, we checked the relevant data for or

ganometallic compounds of U(V). [REDACTED] In $(\text{Cp}^{\text{iPr}_4})_2\text{U}(\mu\text{-N})\text{B}(\text{C}_6\text{F}_5)_3$

(*Chem. Commun.* **56**, 4535-4538 (2020)), the U-Cp distances are in the range of 2.723(3)-2.830(3) Å, and the distances of U-Cp_(cent) are 2.511(1) and 2.520(1) Å. In $\{\text{U}[\eta^8\text{-C}_8\text{H}_6(1,4\text{-Si}(\text{iPr})_3)_2](\text{Cp}^*)(\text{NSiMe}_3)\}$ [REDACTED] (*J. Organomet. Chem.* **857**, 25-33 (2018)), the U-Cp distance is between 2.718(7)-2.866(7), the U-COT distance is between 2.687(6)-2.747(7) Å, and the distances of U-Cp_(cent) are 2.500(1).

[FIGURE REDACTED AT AUTHOR'S REQUEST]

The shortest U-C_{cage} distances in UN@C_s(6)-C₈₂ and UN@C₂(5)-C₈₂ are 2.487(15) and 2.503(7) Å, respectively, similar to the U-Cp_(cent) distances mentioned above. The distances between the metal and the six closest carbons on the fullerene cage are 2.478(15)-2.861(22) Å and 2.503(7)-2.785(7) Å, respectively, which are also close to the U-Cp distances in the abovementioned organometallic compounds. This indicates that the interaction between the fullerene cage and U is likely similar to the coordination between the metal and the cyclopentadienyl group in the organometallic compounds.

In the revised manuscript, we added a comparison with the metal-cyclopentadienyl ligand distances in organometallic compounds after the description of the metal-cage distances in the section '**Molecular and Electronic Structures of UN@C₈₂**'.

7. Line 238. I'm not sure the retention of order of the NiOEP is interesting.

Response: We thank the reviewer for the comments. We revised the sentence as follows: 'the Ni^{II}(OEP) molecule remained completely ordered as the temperature increased from 100 to 273 K' and deleted 'interestingly'.

8. Line 242. Is it helpful to describe the N atom as stationary and U as disordered? The

description on 252 is more useful, as there is still significant order in the U's position and the cage bonding.

Response: We thank the reviewer for this comment. In endohedral metallofullerenes, the nonmetallic atoms tend to be completely ordered, while the metal atoms in many cases are disordered. We note, however, that describing such a sentence here is not significantly helpful for the analysis of the data, so we have removed the corresponding text in Line 242 in the revised manuscript.

9. Line 293. Please define the ratio of d:f in this orbital. How does it compare to the other U-E multiple bonds?

Response: We have added the corresponding ratio of d:f in the text, which was already defined in Figure 4. The d:f ratio is not discussed other than in the context of the self-contained comparison of UN, UN(2+), and the UN clusterfullerenes, and it is not relevant for the bond order analysis provided in the manuscript. The same comment also applies to point 10. By the Reviewer. We added some context and references regarding the bond polarity in response to point 7 of Reviewer #3.

10. Line 313 5f rather than 6d backdonation? Again, please compare the size of the U contributions (4-10 %) to that found in other uranium organometallics such as the Cp and COT sandwiches.

Response: Please see our response to point 9.

11. Line 354 – this is the first time that UC is described as having a genuine triple bond, which I agree with, but which also undermines the statements made in the introduction about the absence of U-E triple bonds in the literature.

Response: We thank the reviewer for this comment. In the introduction we wrote: 'Covalent bonding with the 5f and 6d orbitals in actinide-ligand multiple bonds has been intensively studied, but remains incompletely understood both experimentally and

theoretically'. We assume that the reviewer is referring to this sentence. In this place, we intend to say that the U-E triple bond has been intensively studied but is not fully understood. Ref 60 (*J. Am. Chem. Soc.* **132**, 8484-8488 (2010)) is a combined spectroscopic and theoretical study of UC and CUC, in which the U-C was defined as a well-developed triple bond. However, this U-C triple band has never been discovered in the condensed-phase molecular compound to date. Thus, we think mentioning this work does not seem to contradict the description of the U-E triple bond, which has been intensively studied but is not fully understood.

12. Line 417 please replace 'spins' with a more precise description, including the axis.

Response: We thank the reviewer for the kind suggestion. In the revised version, we revised "the UN spins inside the fullerene cages" to "UN is more mobile and rotates inside the fullerene cage" in the corresponding text.

REVIEWER COMMENTS

Reviewer #1 (Remarks to the Author):

The authors have revised their manuscript and resubmitted.

In the abstract the authors state that actinide diatomics are elusive and only studied in solid inert matrices. They then go on to say they have a diatomic in a fullerene, but it is not a neutral diatomic, it carries a 2+ charge. This is then no different to a coordination complex with a (UN)²⁺ coordinated by one or more ligands. In effect the fullerene is just another ligand (as highlighted by the analogy to Cp and arene binding). It is certainly not the same as a neutral diatomic in a matrix. Given that there are many coordination complexes with diatomic actinide units carrying a variety of charges, since being neutral is obviously not a defining caveat, then it is not objectively credible to say they are only studied in inert matrices. The authors would have a case if the UN were neutral but it is not, they cannot then have their argument both ways. What this then reduces to is that this is the third class of terminal uranium nitride outside of inert matrices.

The authors concede the point that they cannot claim the shortest UN distance. Then they have amended the statement in the abstract to address that point and state that they amended the discussion to “avoid the statement of ‘the shortest’”. However, despite conceding the point around lines 153-156 the authors persist in making shortest statements “Moreover, these UN bonds are shorter than most of the observed UN bond lengths for...”. The first two examples are complexes with UN distances that overlap with the UN fullerene one by the 3sigma criteria.

The authors have provided a range of clarifying statements and extra analysis. I stated before that I was largely convinced of the formulations, details notwithstanding, but my central concerns have not actually been resolved. The authors have made a system in incredibly low yield and quantities, there is limited analysis compared to other systems, in contrast to coordination complexes that are made on higher yields with much more analysis. The authors persist in claiming a shorter/shortest bond when they concede they should not, and they continue to push their novelty arguments too far.

The technical clarifications are appreciated, but a central claim has been conceded and I do not see the novelty or new conceptual insight I expect in a Nature journal so I am unable to support publication here.

Reviewer #2 (Remarks to the Author):

From my point of view all is sorted out now and I recommend publication.

Reviewer #3 (Remarks to the Author):

The article of Chen et al. has been modified after the comments of 4 reviewers and most questions raised were answered. However, in several cases – and with different reviewers, the authors refrained from their original statement of a singular case of UN since many molecular UN features now exist. Thus, I prefer staying on my original statement, which would make this article worthy of publication but in a more specialized journal.

Additionally, on two specific points that were addressed by the authors, I would have more comments:

I am sorry, I am not sure I am satisfied with the answer of the authors on my original point on the different crystallographic positions of the metal center. I agree with the definition, which is made of a disorder as it “reflects the different conformations of the molecule with small energy barriers” but yet the other configuration is still present and thus shall be discussed. Even if the position is “fixed”, meaning that the ellipsoid is of fair size, the second configuration, although minor, should be

considered. I understand that the major is the shortest UN distance but then all following characterization, especially Raman (see comments of Reviewers 1 and 2) should point to both configurations with two distinct UN sets of bands. Is it the case? Although it is a common method used in this peculiar chemistry, I wonder how one can just ignore an existing configuration.

Thanks to the authors for attempting ERP spectroscopy. There are many reasons for having a silent EPR spectrum: the ligand field shall be studied. I wonder if the "shielding effect of the carbon cage" is not a little too vague.

Reviewer #4 (Remarks to the Author):

Overall, I am satisfied with the corrections the authors have made. I am not sure what they can learn from the EPR studies though.

I don't understand why a carbon cage shields an EPR resonance.

The lack of signal could actually tell us about a high symmetry at the U(V) site –

If the symmetry is high enough at the U5+ site, then the lack of an epr signal can indicate that the ground state of the U5+ ion does not contain a $J_z = \pm 1/2$ component, $g(\text{perp})$ can be zero, and thus we get information about the ground state. See the C3v symmetric molecule in

<https://escholarship.org/uc/item/6v96t1kb>

Please could the authors discuss.

Reviewer #1 (Remarks to the Author):

The authors have revised their manuscript and resubmitted.

In the abstract the authors state that actinide diatomics are elusive and only studied in solid inert matrices. They then go on to say they have a diatomic in a fullerene, but it is not a neutral diatomic, it carries a 2+ charge. This is then no different to a coordination complex with a (UN)₂₊ coordinated by one or more ligands. In effect the fullerene is just another ligand (as highlighted by the analogy to Cp and arene binding). It is certainly not the same as a neutral diatomic in a matrix. Given that there are many coordination complexes with diatomic actinide units carrying a variety of charges, since being neutral is obviously not a defining caveat, then it is not objectively credible to say they are only studied in inert matrices. The authors would have a case if the UN were neutral but it is not, they cannot then have their argument both ways. What this then reduces to is that this is the third class of terminal uranium nitride outside of inert matrices.

Response :

(1) We intend to have it “both ways”, in fact. We agree that the interaction between U and fullerene cage can be considered as a *kind of* coordination, as we already wrote in the manuscript. **However, this interaction is notably different from what we know from conventional coordination complexes.** The encapsulated UN unit, analogous to many other endohedral fullerenes (*Chem. Rev.* **113**, 5989–6113(2013); *Chem. Commun.*, **55**, 13000—13003(2019)), has a quite unique interaction with the fullerene cages and can rotate inside the cage at higher temperatures. It is a much ‘softer’ interaction than the coordination between other diatomic units and ligands. Given the strong back-donation from the fullerene cage identified in the computations, the charged species vs. neutral species distinction becomes much less defined than the Reviewer seems to think. Many previous studies, including some of ours, have shown that, due to the unique interactions between the encaged species and fullerene cages, bonding motifs which

have never been observed in conventional coordination complexes can be stabilized inside fullerene cages, which indicates that the fullerene is not just another ligand (e.g., *Nat. Commun.* **9**, 2753 (2018); *J. Am. Chem. Soc.* **141**, 20249-20260 (2019)). Thus, even if the system were considered to be a third class of terminal Uranium nitride, it is very different from the previously reported ones. Therefore, this does not undermine the novelty of this work.

(2) The charged UN unit has similar bonding to the neutral UN, as we already discussed in the manuscript.

(3) To differential the 'neutral' and 'charged' UN units and highlight the unique interaction between fullerene cage and UN unit, we modified the second sentence of the abstract as 'we report a **charged** $U\equiv N$ diatomic species captured in fullerene cages and stabilized by the **unique** U-fullerene coordination.'

The authors concede the point that they cannot claim the shortest UN distance. Then they have amended the statement in the abstract to address that point and state that they amended the discussion to "avoid the statement of 'the shortest'". However, despite conceding the point around lines 153-156 the authors persist in making shortest statements "Moreover, these UN bonds are shorter than most of the observed UN bond lengths for...". The first two examples are complexes with UN distances that overlap with the UN fullerene one by the 3sigma criteria.

Response : To render the statement more precise, we revised the abovementioned text as ' Moreover, these $U\equiv N$ bonds **are relatively short compared to the** observed $U\equiv N$ bond lengths for molecular compounds, such as.....'

The authors have provided a range of clarifying statements and extra analysis. I stated before that I was largely convinced of the formulations, details notwithstanding, but my central concerns have not actually been resolved. The authors have made a system in incredibly low yield and quantities, there is limited analysis compared to other systems, in contrast to coordination complexes that are made on higher yields with

much more analysis. The authors persist in claiming a shorter/shortest bond when they concede they should not, and they continue to push their novelty arguments too far.

Response : We did not intent to state that the novelty of this work is the short/shortest U-N bond, which accordingly is not highlighted in the abstract and conclusion in the revised manuscript. Instead, what we would like to highlight is that such a simple diatomic species can be captured and stabilized inside fullerene cages, forming a new fullerene family with an encaged triple bond. This is a quite unique host-guest molecular structure and indeed unexpected for endohedral fullerene studies. From the perspective of actinide chemistry, the capture of this charged diatomic species in a stable molecular compound provides an unconventional way to study elusive $Ac\equiv E$ bonds, not just limited to $U\equiv N$. We have tried to extend this paradigm to capture other currently elusive but fundamentally important actinide bonding motifs, i.e. $U\equiv C$, by the fullerene cages and preliminary results show that fullerene cages have similar stabilization effect on the elusive $U\equiv C$ bonding motif. We have emphasized this point in the last sentence of abstract and conclusions.

The technical clarifications are appreciated, but a central claim has been conceded and I do not see the novelty or new conceptual insight I expect in a Nature journal so I am unable to support publication here.

Response : It is unfortunate that the Reviewer is so focused on the charged vs. neutral aspect. As already stated, we consider it inappropriate to lump the fullerene cage together with all other ligands, because it offers a unique environment for encapsulated species.

Reviewer #2 (Remarks to the Author):

From my point of view all is sorted out now and I recommend publication.

Response : We thank the Reviewer for the positive evaluation of our work.

Reviewer #3 (Remarks to the Author):

The article of Chen et al. has been modified after the comments of 4 reviewers and most questions raised were answered. However, in several cases – and with different reviewers, the authors refrained from their original statement of a singular case of UN since many molecular UN features now exist. Thus, I prefer staying on my original statement, which would make this article worthy of publication but in a more specialized journal.

Additionally, on two specific points that were addressed by the authors, I would have more comments:

I am sorry, I am not sure I am satisfied with the answer of the authors on my original point on the different crystallographic positions of the metal center. I agree with the definition, which is made of a disorder as it “reflects the different conformations of the molecule with small energy barriers” but yet the other configuration is still present and thus shall be discussed. Even if the position is “fixed”, meaning that the ellipsoid is of fair size, the second configuration, although minor, should be considered. I understand that the major is the shortest UN distance but then all following characterization, especially Raman (see comments of Reviewers 1 and 2) should point to both configurations with two distinct UN sets of bands. Is it the case? Although it is a common method used in this peculiar chemistry, I wonder how one can just ignore an existing configuration.

Response: We thank the reviewer for this suggestion. To address the other configurations corresponds to the minor U sites, in the revised manuscript, we added “ On the other hand, the U-N distances in the other configuration corresponding to its minor U sites are 1.681(7)-1.820(9) Å (U2 (0.1903), U3 (0.1087) and U4 (0.0566)

for UN@C_s(6)-C₈₂) and 1.705(20) Å (U2 (0.188) for UN@C₂(5)-C₈₂) (**Supplementary Fig. 4 and 6 and Supplementary Table 14-15**), all of which are within the bond length range of a U≡N triple bond. ” in the crystal analysis section.

Thanks to the authors for attempting ERP spectroscopy. There are many reasons for having a silent EPR spectrum: the ligand field shall be studied. I wonder if the “shielding effect of the carbon cage” is not a little too vague.

Response: This remark is also raised by Reviewer #4 (see below), who pointed us to check whether the silent EPR feature is due to a ground state Kramers doublet with $g_{\perp} = g_x = g_y = 0$. Analyzing in more detail the electronic structure of UN²⁺ and UN isolated diatomics obtained from the wavefunction theory calculations, we can confirm that both systems exhibit GS Kramers doublets with axial g tensor and $g_{\perp} = g_x = g_y = 0$. **The main manuscript was amended with the following paragraph:** “This behavior is supported by wavefunction theory (WFT) calculations (see Computational Details) on an isolated UN²⁺ diatomic, $d(\text{U-N}) = 1.707 \text{ \AA}$, which predict a $j_z = 5/2$ ground state Kramers doublet (see **Supplementary Table 17**) characterized by an axial g tensor with $g_{\parallel} = g_z = 4.19$ and $g_{\perp} = g_x = g_y = 0$. These values resemble those calculated for a UN²⁺ diatomic with $d(\text{U-N}) = 1.84 \text{ \AA}$ ($g_z = 4.20$, $g_x = g_y = 0$), and both are very close to the expected values for a $|j, j_z\rangle = |5/2, \pm 5/2\rangle$, namely $g_z = 4.29$ and $g_x = g_y = 0$. Similar WFT calculations for an isolated UN diatomic predicted a GS Kramers doublet with $j_z = 7/2$ (see **Supplementary Table 16**) and axial g tensor with $g_z = 3.99$ and $g_x = g_y = 0$, which is an example case for a crystal field GS with $l_z = \pm 5$, $s_z = \mp 3/2$, and $j_z = \mp 5/2$ characterized by $g_z = 4.00$ and $g_x = g_y = 0$. Absence of EPR signals were also reported for (MeC₅H₄)₃UNR compounds where the local C_{3v} symmetry around the metal center renders nil values for g_x and g_y .”

Reviewer #4 (Remarks to the Author):

Overall, I am satisfied with the corrections the authors have made. I am not sure what they can learn from the EPR studies though.

I don't understand why a carbon cage shields an EPR resonance. The lack of signal could actually tell us about a high symmetry at the U(V) site – If the symmetry is high enough at the U⁵⁺ site, then the lack of an epr signal can indicate that the ground state of the U⁵⁺ ion does not contain a $J_z = \pm 1/2$ component, $g(\text{perp})$ can be zero, and thus we get information about the ground state. See the C_{3v} symmetric molecule in <https://escholarship.org/uc/item/6v96t1kb>

Please could the authors discuss.

Response: The reviewer is correct! Please see our response to the last remark of Reviewer #3.

REVIEWERS' COMMENTS

Reviewer #1 (Remarks to the Author):

The authors have presented a revised manuscript. Ignoring shifting priority claims and judging the manuscript in isolation my assessment is that in its current form it is much improved and now strikes the right balance being more precisely worded. It's unfortunate that EPR data are not forthcoming, since they would resolve the fullerene ligand question one way or another since variance, or not, of the g_z value from 4.2 would clarify the 'ligand' effect.

I have only one remaining question, which in the new portion of EPR text the authors state "These values resemble those calculated for a UN₂⁺ diatomic with $d(\text{U-N}) = 1.84 \text{ \AA}$ ($g_z = 4.20$, $g_x = g_y = 0$)," . What are they referring to, a prior study?

Reviewer #2 (Remarks to the Author):

After carefully reading and considering the comments of the other reviewers, I recommend publication in a more specialized journal.

Reviewer #3 (Remarks to the Author):

I sincerely appreciate the efforts made by the authors to answer all my technical comments during this review process. Although I am satisfied with these, I am still finding that the work presented herein is not suitable for Nature Communications because I don't see the findings of exceptional novelty. As stated in my two previous reviews, I'll find the work more suitable for a more specialized journal.

Reviewer #1 (Remarks to the Author):

The authors have presented a revised manuscript. Ignoring shifting priority claims and judging the manuscript in isolation my assessment is that in its current form it is much improved and now strikes the right balance being more precisely worded. It's unfortunate that EPR data are not forthcoming, since they would resolve the fullerene ligand question one way or another since variance, or not, of the g_z value from 4.2 would clarify the 'ligand' effect.

I have only one remaining question, which in the new portion of EPR text the authors state "These values resemble those calculated for a UN_2^+ diatomic with $d(U-N) = 1.84 \text{ \AA}$ ($g_z = 4.20$, $g_x = g_y = 0$)," . What are they referring to, a prior study?

Response : We thank the reviewer for this question. Here we missed a citation and have added it in the corresponding text (*Nat. Commun.* **7**, 13773 (2016)). It has been cited elsewhere as ref. 22. For details please see the revised manuscript.